# Assessing the climate and air quality effects of future aerosol mitigation in India using a global climate model combined with statistical downscaling

Tuuli Miinalainen[1,2], Harri Kokkola[3], Antti Lipponen[3], Antti-Pekka Hyvärinen[4], Vijay Kumar Soni[5], Kari E. J. Lehtinen[1,3], and Thomas Kühn[1,3,6]

[1]Department of Technical Physics, University of Eastern Finland (UEF), Kuopio, Finland
[2]Earth Observation Research, Finnish Meteorological Institute (FMI), Helsinki, Finland
[3]Atmospheric Research Centre of Eastern Finland, Finnish Meteorological Institute (FMI), Kuopio, Finland
[4]Atmospheric Composition Research, Finnish Meteorological Institute (FMI), Helsinki, Finland
[5]India Meteorological Department (IMD), Ministry of Earth Sciences, New Delhi, India
[6]Weather and Climate Change Research, Finnish Meteorological Institute (FMI), Helsinki, Finland

**Correspondence:** Tuuli Miinalainen (tuuli.miinalainen@uef.fi)

**Abstract.** We studied the potential of using machine learning to downscale global-scale climate model output towards ground station data. The aim was to analyze simultaneously both city-level air quality and regional and global scale radiative forcing values for anthropogenic aerosols. As the city-level air pollution values are typically underestimated in global-scale models, we used a machine learning approach to downscale fine particulate (PM2.5) concentrations towards measured values. We first simulated the global climate with the aerosol-climate model ECHAM-HAMMOZ, and corrected the PM2.5 values for the Indian mega-city New Delhi.

The downscaling procedure clearly improved the seasonal variation of the model data. The seasonal trends were much better captured in the corrected PM2.5 than in original ECHAM-HAMMOZ PM2.5 when compared to the reference PM2.5 from the ground stations. However, short-term variations showed less extreme values with the downscaling approach. We applied the downscaling model also to simulations where the aerosol emissions were following two different future scenarios: one following the current legislation and one assuming currently maximum feasible emission reductions. The corrected PM2.5 concentrations for the year 2030 showed that mitigating anthropogenic aerosols improves local air quality in New Delhi, with organic carbon reductions contributing most to these improvements.

In addition, aerosol emission mitigation also resulted in negative radiative forcing values over most of India. This was mainly due to reductions in absorbing black carbon emissions. For the two future emission scenarios modelled, the radiative forcing due to aerosol-radiation interactions over India was $(-0.09\pm0.26)\,\mathrm{W\,m^{-2}}$ and $(-0.53\pm0.31)\,\mathrm{W\,m^{-2}}$, respectively, while the effective radiative forcing values were $(-2.1\pm4.6)\,\mathrm{W\,m^{-2}}$ and $(0.06\pm3.39)\,\mathrm{W\,m^{-2}}$, respectively. Although accompanied with relatively large uncertainties, the obtained results indicate that aerosol mitigation could bring a double benefit in India: better air quality and decreased warming of the local climate.

Our results demonstrate that downscaling and bias correction allow more versatile utilization of global-scale climate models. With the help of downscaling, global climate models can be used in applications where one aims to analyze both global and regional effects of policies related to mitigating anthropogenic emissions.

*Copyright statement.* TEXT

## 1 Introduction

The climate crisis and air quality issues are strongly interlinked. Air pollutants have diverse characteristics and hence each pollutant has a different impact on air quality and on atmospheric processes. Furthermore, many pollutants are interlinked, and decreasing emissions of one pollutant might reduce also the emissions of co-emitted pollutants, which affects the net impact of the total emission reduction. Therefore, improving air quality can bring either co-benefits or trade-offs when aiming to slowing down global warming. Analyzing the effects of emission mitigation on air quality and on global climate are usually done using

separate tools from different model families (Trail et al., 2013; Stohl et al., 2015; Gao et al., 2018): Air quality models usually operate on small domains and in the lower part of the atmosphere, but have relatively high horizontal and vertical resolution, while climate models operate on large to global scales, but use much coarser horizontal and vertical resolutions.

 Fine particulate matter (PM2.5) air pollution is associated with millions of premature deaths globally each year (Burnett et al., 2018; Vohra et al., 2021) and is one of the most significant causes of global disease burden (GBD 2015 Risk Factors

Collaborators, 2016). During the past decades, the global trends for PM2.5 have been slightly increasing (Hammer et al., 2020). Half of the global population are exposed to increasing levels of air pollution (Shaddick et al., 2020), despite the decreasing trends in Europe and North America (Hammer et al., 2020). PM2.5 refers to particulate matter (PM) which is composed of either solid or liquid atmospheric particles with diameters of less than $2.5\,\mu\mathrm{m}$. Particulate matter is a mixture of various chemical species such as sulfate, organic carbon (OC) and black carbon (BC), and is formed due to both human activities and

natural processes. For instance, residential biomass burning, road transport, agricultural activities and industrial operations are common sources of atmospheric PM2.5.

 Besides health effects, PM aerosols affect atmospheric processes and Earth's energy balance via various mechanisms. Some of the substances contained in aerosol particles, for instance OC and sulfate, generate a negative radiative forcing relative to pre-industrial times due to scattering of shortwave (SW) radiation. On the other hand, BC-containing aerosol particles can absorb

SW radiation, resulting in a positive radiative forcing. These are termed aerosol-radiation interactions (ARI). Furthermore, aerosol particles can act as seeds for cloud droplets and, hence, changes in aerosol composition and concentration can alter cloud properties, affecting Earth's radiative balance indirectly (aerosol-cloud interactions; ACI). Aerosols can also affect local meteorological dynamics. For instance, absorbing aerosol particles can alter atmospheric stability in the troposphere due to local heating (Johnson et al., 2019). The combined radiative effect of aerosols is still highly uncertain, but is estimated to lead

to a net cooling effect (Bellouin et al., 2020). Mitigation of short-lived climate forcer (SLCF) emissions, especially BC, has

been considered as one potential pathway to slow down global warming, as the lifetime of these pollutants is short compared to the main greenhouse gas (GHG) carbon dioxide ($CO_2$). However, some studies also found that, due to ACI and reductions of associated co-emitted, climate-cooling pollutants, mitigation of BC emissions may end up warming the climate, or, at least the overall effect includes significant uncertainty (Cherian et al., 2017; Kühn et al., 2020; Harmsen et al., 2020).

Atmospheric general circulation models (GCMs) and Earth system models (ESMs) are designed to study large-scale or global climate effects and hence operate on relatively coarse horizontal ($0.5°$–$2.0°$) and vertical resolutions. They are therefore not optimal tools to accurately model surface PM2.5 concentrations that correspond to exact point measurements (e.g. in cities). Hence, the regional characteristics affecting local air pollution levels are many times not captured (Li et al., 2016; Cooney, 2012; Turnock et al., 2020). Moreover, the input data describing both anthropogenic and natural aerosol emissions might

lack information of some local sources (Kukkonen et al., 2018) or overestimate emissions compared to real-life conditions (Wang et al., 2021), which directly affects the model estimates for PM2.5. Turnock et al. (2020) concluded that the Coupled Intercomparison Model Project 6 (CMIP6) models underestimated the surface PM2.5 by up to $10\,\mu g/m^3$, and that models tend to produce different regional patterns due to differences in chemistry and aerosol processes. Air quality models, on the other hand, are not well suited to assess climate impacts of emission changes. Furthermore, when modelling future emission

scenarios, air quality models require input from global climate models to set boundary and initial conditions.

    The advantage of using a GCM or ESM is that air quality and climate impacts can be studied at the same time. Furthermore, air quality effects can be analyzed on the entire model domain, which may help lowering the computational cost of the assessment, especially when assessing the effects of future emission changes. In order to remedy the resolution-related bias problems described above, statistical downscaling can be a viable alternative to higher resolution physical modelling in these scenarios.

There exist many statistical downscaling methods for correcting GCM-derived data (Ivatt and Evans, 2020; Geng et al., 2020; Nolte et al., 2018; Lu and Wang, 2005; Lipponen et al., 2013). In contrast to dynamical downscaling (Nolte et al., 2018), a benefit of statistical downscaling is that data from a coarse resolution model can be corrected after a simulation (offline) using measured data or high resolution model data. The advantage of such an approach is that the bias can be corrected without having to, e.g., derive, new parameterizations or implementing new in-model methods for grid refinement. In addition,

statistical downscaling is computationally inexpensive and often faster than dynamical downscaling methods (Tran Anh and Taniguchi, 2018). Various machine learning approaches have been suggested for downscaling or predicting air pollution levels (Ivatt and Evans, 2020; Geng et al., 2020; Zamani Joharestani et al., 2019; Silibello et al., 2015; Watson et al., 2019; Yang et al., 2020).

    In this study, we investigated the potential of using machine learning to improve PM2.5 concentrations derived with a GCM.

We simulated the PM2.5 fields with the aerosol-climate model ECHAM-HAMMOZ, and corrected the data afterwards with a downscaling approach, limiting the correction to PM2.5 concentrations modelled for the Indian mega city of New Delhi. The aim of this work is to explore more versatile methods to utilize global model data. The combination of ECHAM-HAMMOZ and statistical downscaling could be used as a relatively light-weight tool for applications where one aims to assess the effects of aerosol mitigation simultaneously on larger scale climate, and on local air quality.

Here our focus is on India and specifically the New Delhi region. India, overall, is one of the countries with the highest BC emissions globally (Xu et al., 2021) and emissions originating from the Indian subcontinent contribute significantly to the Asian Tropopause Aerosol layer (ATAL); (Fairlie et al., 2020; Lau et al., 2018) which has been suggested to affect the regional surface temperatures and radiative forcing over Asia during the past decade (Vernier et al., 2015). Despite the various programs tackling air pollution and, partly, due to the polycentric nature of environmental regulation in India (Honkonen, 2020), many

Indian cities still struggle with high PM2.5 concentrations at alarming levels. The premature mortality due to PM in India is estimated to be over a million cases annually (Vohra et al., 2021; Guo et al., 2018; Sahu et al., 2020). The aerosol emissions from South Asia also affect Arctic black carbon concentrations, which directly links to Arctic warming (Backman et al., 2021; Zhao et al., 2021). Thus New Delhi and its surroundings is a very interesting region to test if there are co-benefits or trade-offs with regard to climate and air quality when reducing BC emissions along with the co-emitted pollutants. In addition, New

Delhi PM2.5 shows strong seasonal variation and high dependence on anthropogenic emissions (Bawase et al., 2021). This makes New Delhi an ideal target region to study how well downscaling can capture short- and long-term trends in PM2.5.

In this article, we employed random forest regression for downscaling global model-derived surface PM2.5 concentrations such that the regression algorithm corrected the biases between measured and modeled PM2.5 values at individual stations. In addition, we applied the trained random forest function to three separate climate model simulations which were conducted

with present-day and future aerosol emission projections for the years 2015 and 2030, respectively. We used the ECLIPSE V6b emission scenarios, which project global emission changes for the current legislation (CLE) (Stohl et al., 2015) and for maximum feasible global mitigation of SLCFs (Im et al., 2021). Our aim was to simultaneously study the effects of global SLCF mitigation on Earth's energy balance and on air quality. The air quality assessment was limited to the city of New Delhi, but could easily be extended using additional observational data.

## 2   Methods

### 2.1   Aerosol-climate model ECHAM-HAMMOZ

We performed all the global model simulations with the aerosol-climate model ECHAM6.3-HAM2.3 (ECHAM-HAMMOZ) (Schultz et al., 2018). ECHAM-HAMMOZ consist of the general circulation model ECHAM (Stevens et al., 2013), the aerosol module HAM (Kokkola et al., 2018; Tegen et al., 2019; Neubauer et al., 2019), and, in our setup, the aerosol microphysics

module SALSA2.0 (Kokkola et al., 2018). HAM treats the chemical compounds black carbon (BC), organic aerosol (OA), sulfate (SU), mineral dust (DU) and sea salt (SS). SALSA discretizes the aerosol size distribution into 10 size classes and separately treats a soluble and an insoluble sub-population. A detailed description of SALSA is presented in Kokkola et al. (2018). SALSA has been further developed to include an improved wet-scavenging scheme, where fixed scavenging coefficients were replaced by an algorithm that takes into account the fraction of activated particles (Holopainen et al., 2020). In

this study, we used T63L47 grid resolution, which corresponds to an approximately $1.9\,° \times 1.9°$ horizontal resolution, and has 47 vertical hybrid layers up to $0.01\,\mathrm{hPa}$ (appr. $80\,\mathrm{km}$) altitude. In addition, we used an additional setup where the emitted BC

**Table 1.** ECLIPSE V6b emissions, global yearly sums for current legislation (CLE) scenario emissions for the years 2015, 2020 and 2030 and for maximum feasible reductions (MFR) scenario for the year 2030. The second column shows the global summed emitted mass, and the third column represents the summed emitted mass for New Delhi surroundings (24–34° N, 72–82° W).

| emission scenario | Global sum | | | New Delhi surroundings | | |
|---|---|---|---|---|---|---|
| | BC (kt yr$^{-1}$) | OC (kt yr$^{-1}$) | SO$_2$ (kt yr$^{-1}$) | BC (kt yr$^{-1}$) | OC (kt yr$^{-1}$) | SO$_2$ (kt yr$^{-1}$) |
| CLE, year 2015 | 6351.5 | 13 763.2 | 73 335.1 | 295.2 | 665.3 | 1995.0 |
| CLE, year 2020 | 5909.1 | 13 595.4 | 55 059.6 | 254.9 | 671.6 | 2012.7 |
| CLE, year 2030 | 5378.9 | 13 752.9 | 47 503.9 | 231.3 | 711.4 | 1413.8 |
| MFR, year 2030 | 1985.2 | 3492.2 | 21 673.9 | 78.4 | 133.8 | 548.4 |

was assumed to get directly internally mixed with sulfate (Holopainen et al., 2020). Therefore, BC-containing particles were modeled to be more soluble already at the time of their emission.

## 2.2 Anthropogenic aerosol emissions

For the anthropogenic aerosol emissions in ECHAM-HAMMOZ, we used the ECLIPSE V6b emission inventory (Stohl et al., 2015; Klimont et al., 2017; IIASA, 2021; Im et al., 2021; AMAP, 2015) which was compiled with the integrated assessment model GAINS (Amann et al., 2011; Klimont et al., 2017). For this study, we used only the emissions of BC, OC and sulfur dioxide (SO$_2$), and re-gridded the emission fields to correspond to the spatial grid resolution T63 used in ECHAM-HAMMOZ ($\sim 2° \times 2°$). The global emissions and the regional emissions near New Delhi from the ECLIPSE V6b inventory are presented in Table 1.

The current legislation (CLE) scenario (Belis et al., 2022) projects a 15 % decrease in global BC emissions between 2015 and 2030 and a 35 % decrease in SO$_2$ emissions. The change in net global OC emissions is almost negligible ($\sim -0.07$ %) due to increasing emissions in the waste and agricultural waste burning sectors, which compensate emission reductions projected in the domestic and traffic sectors. In the area surrounding New Delhi, the decrease in BC and SO$_2$ is approximately at the same level as for the global sum: BC emissions decrease by 22 % between 2015 and 2030 for the CLE scenario, and SO$_2$ emissions are 29 % smaller for 2030. The OC emissions are projected to increase for the New Delhi surrounding by 7 % in the year 2030 compared to the year 2015. This is due to increasing emissions from the waste and industry sectors, despite emission cuts projected for the domestic and traffic sectors.

The maximum feasible reductions (MFR) scenario is built on assumptions where the most advanced SLCF emission reduction technologies that are included in the GAINS model are implemented fully globally (Belis et al., 2022; Im et al., 2021). In MFR 2030 the global BC, OC and SO$_2$ emissions decrease by 69 %, 75 % and 70 %, respectively, compared to CLE 2015. The corresponding decreases for the New Delhi surroundings are 73 %, 80 % and by 73 %, respectively.

Both globally and in the New Delhi surroundings, the sectors contributing most to reductions in BC emissions were the domestic and traffic sectors, while for OC, the domestic and waste sectors had the most significant reductions. For SO$_2$, the largest reductions were projected in the MFR scenario for the energy and industry sectors.

## 2.3 Model simulation setup

We conducted altogether four simulations with ECHAM-HAMMOZ: for teaching the random forest algorithm, and further applying the correction to simulations with present-day and future aerosol emission scenarios. The simulations are listed in Table 2.

| Simulation | Meteorology | Aerosol Emissions | time |
|------------|-------------|-------------------|------|
| RF_TRAIN | nudged, ERA5 | ECLIPSE V6b CLE yearly interpolated emissions | 2016–2019 |
| PRES | freely evolving | ECLIPSE V6b CLE 2015 repeated | 10 simulation years |
| CLE_2030 | freely evolving | ECLIPSE V6b CLE 2030 repeated | 10 simulation years |
| MITIG_2030 | freely evolving | ECLIPSE V6b MITIG 2030 repeated | 10 simulation years |

**Table 2.** Experiment design

The first simulation, (**RF_TRAIN**) was used for training and validating the random forest regression model in present day conditions. For the RF_TRAIN, we used the Newtonian relaxation scheme (nudging) for large scale meteorological fields. Following the recommendation by Zhang et al. (2014), we nudged the wind patterns and surface pressure towards ERA5 reanalysis data (Hersbach et al., 2020; Copernicus Climate Change Service (C3S), 2017), and allowed temperature and free static energy to evolve freely. For the anthropogenic aerosol emissions, we used the ECLIPSE V6b current legislation (CLE)

scenario and calculated the yearly emissions by linearly interpolating the values for each grid box and emission species from year 2015 to year 2020. The monthly values were then computed based on the yearly values and the monthly pattern from the ECLIPSE inventory. RF_TRAIN was integrated between the years 2016 and 2019 with an output time resolution of 3 hours.

The rest of the simulations (i.e. **PRES**, **CLE_2030** and **MITIG_2030**) were used to simultaneously analyze radiative forcings and local air quality in New Delhi under two different future emission scenarios, CLE and MFR, with freely evolving

wind and pressure fields, i.e. no nudging was applied. For each of these simulations, the PM2.5 values were bias-corrected using the RF model which was trained with RF_TRAIN. In addition to the PM2.5 fields, we computed the corresponding global radiative forcing values to see how the changes in air pollutants affect the net radiation at the top of the atmosphere. The additional simulation PRES, which was used as a reference simulation to describe the present day air pollution conditions, was necessary, because the radiation fields from the nudged simulation RF_TRAIN would not be compatible with the radiation

fields from CLE_2030 and MITIG_2030.

In Simulation PRES, for anthropogenic aerosol emissions we used the ECLIPSE V6b CLE emissions for the year 2015, repeating the monthly emissions for 10 simulation years. Similarly, the simulation **CLE_2030** used ECLIPSE V6b CLE scenario and had 10 simulation years, but the anthropogenic aerosol emissions were for the year 2030. Simulation **MITIG_2030** was identical to CLE 2030, but instead of CLE scenario, we used the global MFR scenario with emissions for the year 2030.

Sea surface temperature (SST) and sea ice cover (SIC) were fixed to prescribed fields from the Program for Climate Model Diagnosis & Intercomparison's (PCDMI's) Atmospheric Model Inter-comparison Project (AMIP) data (Taylor et al., 2012).

For Simulation RF_TRAIN, we used the monthly mean values for each year, except for year 2019, where we used 2018 values since the data were not available by the time the simulations were done. For the rest of the simulations, we used the mean climatological values for each month, calculated from the monthly mean data between the years 2000 and 2015.

In all of the simulations, sea salt and dust emissions were calculated online, and were dependent on the 10 meter wind speed (Tegen et al., 2019). The calculation routine for dust emissions used the parameterization of Tegen et al. (2002) that has been further improved by Cheng et al. (2008) and Heinold et al. (2016). The aerosol emissions for aviation were kept fixed for all of the simulations, repeating the monthly mean emissions for the year 2015 with the Representative Concentration Pathway (RCP) scenario 4.5 (Thomson et al., 2011; van Vuuren et al., 2011) from the Emissions for Atmospheric Chemistry and Climate

Model Intercomparison Project (ACCMIP) database (Lamarque et al., 2010). In addition, the hydroxyl radical (OH) mixing ratios were taken from reanalysis data as described in Inness et al. (2013). For ozone mixing ratios, we used Chemistry-Climate Model Initiative (CCMI) data that were prepared for the CMIP6 simulations (Hegglin et al., 2016). In the simulations PRES, CLE_2030 and MITIG_2030, we used historical monthly varying climatologies that were calculated based on monthly mean values between years 2000 and 2014. With the simulation RF_TRAIN, we used monthly mean values for each simulation year,

following the Shared Sosioeconomic Pathway (SSP) scenario 2 (Fricko et al., 2017). Forest fire emissions were taken from the CAMS Global Fire Assimilation System (GFAS) emission inventory (Kaiser et al., 2012). For the simulation RF_TRAIN, we used the daily mean emissions for each simulation year. For the rest of the simulations, we computed the monthly varying climatological values that were calculated based on the GFAS data between the years 2000 and 2016.

## 2.4   Random forest regression

Random forest (RF) (Ho, 1995; Breiman, 2001) is a supervised ensemble learning method which aggregates multiple decision trees (Blockeel and De Raedt, 1998). The output of the algorithm is the mean value of the predictions of all the trees. This helps to avoid overfitting, which is a typical problem for predictions with a single regression tree (Bramer, 2013). Therefore, RF is suitable for modelling complex and non-linear relationships (Auret and Aldrich, 2012). Decision trees are used in statistical modeling for both classification and regression. A decision tree uses the classification and regression trees (CART) algorithm

(Breiman et al., 1984), where the input data space is partitioned down into subspaces. In the case of regression, the mean square error of the data is typically used as a measure of homogeneity of each subspace (Breiman, 2001). RF was shown to be highly effective in a variety of applications (Fawagreh et al., 2014), and has also been widely utilized in the field of climate modelling (Crawford et al., 2019; Lipponen et al., 2013; Pang et al., 2017).

    In this study, we used the RF regression method from the Scikit Learn Python module (Pedregosa et al., 2011). Based on

testing different combinations, our RF model setup was not very sensitive to the choice of hyper parameters. Furthermore, we aimed to have the same hyper parameters in all of the individual RF models (see Section 2.6). Therefore, we used the recommended default values for the hyper parameters. There was no maximum depth for each tree set and the depth of each tree was determined automatically based on the splitting criterion. The number of trees was the default value, 100. The splitting criterion was also set according to the default settings, i.e. the mean squared error. The maximum number of input features to

be used in one decision tree node (max_features) was set to 7, which corresponds to the recommended one third of the total

number of input features (Trevor et al., 2009). We evaluated the importance of each input feature by using a built-in function from the Scikit-Learn Random Forest regressor module (Pedregosa et al., 2011). The importance for each input feature is calculated as the total decline in the splitting criterion due that specific input feature. The final feature importance values are then normalized to an interval between 0 and 1.

## 2.5  Measurement station data

RF was applied to correct the modeled ECHAM-HAMMOZ PM2.5 data to correspond to those measured in 31 ground stations located in the New Delhi capital region. The ground measurement station network in New Delhi is extensive and there was a good amount of measurement data available, which made New Delhi ideal location for this kind of study.

We used data from each station to train and validate RF models, which were then applied to three climate model simulations. A list of the stations and their spatial coordinates are presented in Table S1 and in a map in Figure S1. The stations are administrated by three different operators. The operator for each station is also marked in Table S1.

Delhi Pollution Control Committee (DPCC) (Delhi Pollution Control Committee, 2022) operates 18 out of the 31 stations from which data was retrieved for this study. We also harnessed PM2.5 data from seven "SAFAR" stations that are administered by the India Meteorological Department (IMD) (System of Air Quality and Weather Forecasting And Research, 2022). In addition, we used measurement data from six stations operated by the Central Pollution Control Board (CPCB). PM2.5 mass concentration is measured at all the stations using Beta Attenuation Monitors (BAMs) employing the $\beta$-ray attenuation technique. It is a United States Environmental Protection Agency (US EPA) recommended method and adopted at all the continuous ambient air quality monitoring stations in Delhi (CPCB, 2020; Saraswati et al., 2019; Hama et al., 2020; Sharma et al., 2022). The lower detection limit of monitors is $0.1\,\mu g\,m^{-3}$. Operating bodies frequently standardize the monitors as per the CPCB guidelines for verifying the quality of the data.

The stations are spread to a square area of approximately $30\,km \times 30\,km$, and data were provided for a time period of 1.1.2016–31.12.2019. There are some differences in temporal data coverage between the stations. The data coverage percentage for each year and station is reported in Table S1. For the year 2016, there are only five stations that have over $50\,\%$ time coverage. The same applies for the year 2017. However, there is over $80\,\%$ coverage for the years 2018 and 2019 for almost all of the stations. These differences in temporal coverage affect the RF downscaling results: the final RF bias correction leans more on the year 2018 data since there is a higher number of measurement points for that year.

The original data for the stations had a time resolution of $30\,min$. We calculated the daily average PM2.5 concentration values for all of the stations, and used those in the RF model training. Initially there were data available for almost 40 ground measurement stations. However, some of the station data series did not include sufficient amount of data for the random forest training phase. Therefore, we excluded the stations from the original set, which had less than 180 days of PM2.5 concentration data for the training phase from our analysis, which left us with the 31 stations discussed above. All in all, we had good temporal coverage for the measurement station data for both training and testing phases.

## 2.6 Developing the RF model for bias correction

Instead of implementing the RF model to predict the PM2.5 concentrations directly, we used the model data to compute the bias $\epsilon$ between the measured and modelled PM2.5 values and thereby predict the bias between modelled and actual PM2.5 concentrations. This approach has the advantage of better incorporating the modelled physical information into the training and prediction phases leading to overall more accurate results (Lipponen et al., 2013). Predicting the bias instead of the absolute value also helps minimizing problems with predictions outside of the training data.

We divided the RF_TRAIN and ground measurement station data into training phase data (train) and testing phase data (test). We used the first two-thirds of the RF_TRAIN simulation data for training the RF model, which corresponds to the time interval between 1.1.2016 and 21.8.2018. The rest of the data (corresponding to 22.8.2018–31.12.2019) was used for testing the RF model. We intentionally used separate time spans for the training and testing in order to obtain more realistic estimates for the accuracy or our method. If we would have randomly divided all the data into training and testing sets, the likelihood to have very similar samples in both data sets would have been much larger than in our approach, leading to overoptimistic results.

We conducted the random forest correction in the following way: At first, we calculated the absolute bias $\epsilon_{\text{train}}$ between each measurement station and ECHAM-HAMMOZ-derived PM2.5 without the mineral dust component (PM25_no_DU). The mineral dust component was excluded due to the temporal differences in dust episodes: the PM2.5 concentrations peaks in ECHAM-HAMMOZ did not appear exactly for the same dates as in the measurements, and this could induce artificially large bias values when training the RF. Furthermore, due to dust episodes, the maximum PM2.5 for ECHAM-HAMMOZ was during June and July, whereas measurement stations showed maxima mostly for the months of December and January. However, mineral dust was included in the input training set (see Table 3), i.e., dust can affect the RF bias correction. PM2.5 values from ECHAM-HAMMOZ which exclude the mineral dust component are hereafter referred to as non-dust PM2.5. Then, we trained a RF model with the input feature data set $x_{\text{train}}$ (see Table 3), and set the bias $\epsilon_{\text{train}}$ as a target for training.

$$f = \text{RANDOM\_FOREST}(x_{\text{train}}, \epsilon_{\text{train}}) \tag{1}$$

For easier statistical analysis, this was done separately for each station. Note that many stations did not provide complete data for the whole training phase period. In case of missing data, we removed those samples from the input feature data $x_{\text{train}}$ that were not present in the measurement station data.

After that, we calculated a RF prediction for the bias using the test phase input feature data ($x_{\text{test}}$), using the RF model trained for one individual station. Then we computed the corresponding PM2.5 value by correcting the ECHAM-HAMMOZ PM2.5 with

$$\text{PM2.5}_{\text{RF prediction}} = \text{PM2.5}_{\text{ECHAM-HAMMOZ,no dust,test}} + f(x_{\text{test}}) \tag{2}$$

**Table 3.** Input features used in the random forest fitting. The corresponding abbreviations used in Figures 1 and 2 are written in brackets. Feature data is retrieved only for one grid box (point), or over a larger area (24–34° N, 72–82° W). For the larger area, we either calculated the area-weighted mean value (fldmean) or the sum over the whole area (fldsum). Burden indicates vertically integrated concentration values.

| variable | point, fldmean or fldsum | variable | point, fldmean or fldsum |
|---|---|---|---|
| PM2.5 (PM25) | point | 10m wind speed (ws10) | fldmean |
| PM2.5 without mineral dust (PM25_no_DU) | point | sin(10m wind direction) (wd10_sin) | fldmean |
| PM2.5 due to BC (PM25_BC) | point | cos(10m wind direction) (wd10_cos) | fldmean |
| PM2.5 due to sulfate (PM25_SO4) | point | mineral dust burden (burden_DU) | fldmean |
| PM2.5 due to sea salt (PM25_SS) | point | BC burden (burden_BC) | fldmean |
| PM2.5 due to mineral dust (PM25_DU) | point | SO4 burden (burden_SO4) | fldmean |
| PBL pressure (pbl_p) | point | mineral dust total emission (emi_DU) | fldsum |
| temperature at 2m (temp2) | point | BC total emission (emi_BC) | fldsum |
| large scale precipitation (aprl) | fldmean | OC total emission (emi_OC) | fldsum |
| convective precipitation (aprc) | fldmean | $SO_2$ total emission (emi_SO2) | fldsum |

The final prediction for RF-corrected New Delhi PM2.5 concentration was then formed as the average of the individual RF predictions that were computed for each, station-specific RF model. The error statistics were calculated between the daily average prediction and the daily average of all measurement stations. The input features used in the random forest fitting are purely ECHAM-HAMMOZ data, and we did not use any external data as input predictors. The input features are presented in Table 3.

All of the variables presented in Table 3 are post-processed output from ECHAM-HAMMOZ simulations. We selected the input features mainly based on the average feature importance values. At first, we formed a larger set of input features and ran the RF modelling with this larger set of input features. After that, we pruned out those variables that had an average feature importance value close to zero, i.e., they were assumed to have a very minor influence on the RF correction. Finally, we identified pairs of strongly correlated input features (with correlation coefficients larger than 0.9) and removed the one of the two features that had a smaller average feature importance value.

For some input features, we used values representing one grid box surrounding the New Delhi region (point). These were retrieved from ECHAM-HAMMOZ data by using nearest neighbor interpolation. For the rest of the features, we first defined a square of 5x5 grid boxes surrounding New Delhi (24–34° N, 72–82° W). For the variables describing emissions, we computed the daily total emitted mass ('fldsum' in Table 3) over the whole square. For all other non-point variables, we calculated an area-weighted mean value ('fldmean' in Table 3) for each time step and feature. In the end, we calculated the daily average values for all input features.

## 2.7 Employing the RF model to free-running simulations

We implemented the RF model further to simulations PRES, CLE_2030 and MITIG_2030. This was done in order to estimate simultaneously local PM2.5 concentrations in New Delhi as well as regional and global radiative effects of two different future aerosol emission scenarios. Simulations PRES and RF_TRAIN were also used to evaluate, how much nudging affects the RF correction results. One should keep in mind that in this bias correction approach, the models are assumed to be city specific. In other words, RF models trained with New Delhi data are not meant for correcting biases in ECHAM-HAMMOZ PM2.5 for other cities.

Again, we constructed an individual RF model for each measurement station, and then calculated the final prediction as an average of multiple models. For training each model, this time we employed the whole RF_TRAIN input data set, i.e. all data from the period of 1.1.2016–31.12.2019, and chose only the time steps that were available for each of the measurement stations. Then we applied the trained RF model to the simulations PRES, CLE_2030 and MITIG_2030 to get the bias prediction for PM2.5, and computed the corrected PM2.5 concentrations. Finally, we calculated the multi-year average daily values.

## 2.8 Calculating radiative forcings

For analyzing the impacts of aerosol emission mitigation on global warming, we computed the global radiative forcing values for CLE_2030 and MITIG_2030 simulations. The radiative effect due to aerosol-radiation interactions ($RE_{ARI}$) is computed in ECHAM-HAMMOZ according to Collins et al. (2006). The $RE_{ARI}$ is the difference in radiative flux with and without aerosol, and is calculated with a double-call to the radiation routine (Collins et al., 2006). The radiative forcing due to aerosol-radiation interactions ($RF_{ARI}$) is then calculated as the difference in $RE_{ARI}$ between a perturbed simulation (here CLE_2030 or MITIG_2030) and the reference simulation (PRES) (Ghan, 2013; Neubauer et al., 2019). It should be noted that the cloud properties between the perturbed and reference simulations are not identical, and therefore, the $RF_{ARI}$ calculated here does not fully correspond the $RF_{ARI}$ defined in Myhre et al. (2013). However, these changes in cloud conditions are estimated to have a relatively small effect on $RF_{ARI}$ (Neubauer et al., 2019).

We furthermore calculated the total effective radiative forcing ($ERF_{ARI+ACI}$), which includes $RF_{ARI}$ as well as radiative forcings due to aerosol-cloud interactions ($RF_{ACI}$) and rapid adjustments (Boucher et al., 2013; Forster et al., 2021). Here we computed the $ERF_{ARI+ACI}$ as the difference in the net radiative flux at the top of the atmosphere (TOA) between a perturbed scenario and the reference simulation (Forster et al., 2021), using simulations with fixed SST and SIC. For simplicity, we denote the $ERF_{ARI+ACI}$ here as ERF.

In order to compute $RF_{ARI}$ and ERF, we first computed 2D yearly mean fields of $RE_{ARI}$ and the net TOA radiative flux, respectively, which were calculated as yearly mean values for each separate grid box. In addition, we also computed area-weighted average values over India and over the entire globe. After that, we calculated the 10 year average for each simulation, and similarly, the standard deviation of the yearly mean values. Finally, the $RF_{ARI}$ and ERF values were computed as the difference between CLE_2030 and PRES, and similarly between MITIG_2030 and PRES, as described above. The combined standard deviation was estimated as $\sigma = \sqrt{\sigma_{\text{pert}}^2 + \sigma_{\text{reference}}^2}$.

# 3 Results

## 3.1 Downscaling PM2.5 concentrations with random forest regression

315 We first trained the RF model using RF_TRAIN and measurement station data for 1.1.2016–21.8.2018 as discussed in Section 2.6. Then, we tested the model by applying the trained RF model to the latter part of the RF_TRAIN data, 22.8.2018–31.12.2019. The results of the RF correction to the testing phase, and the data used in the training phase, are presented in Fig. 1.

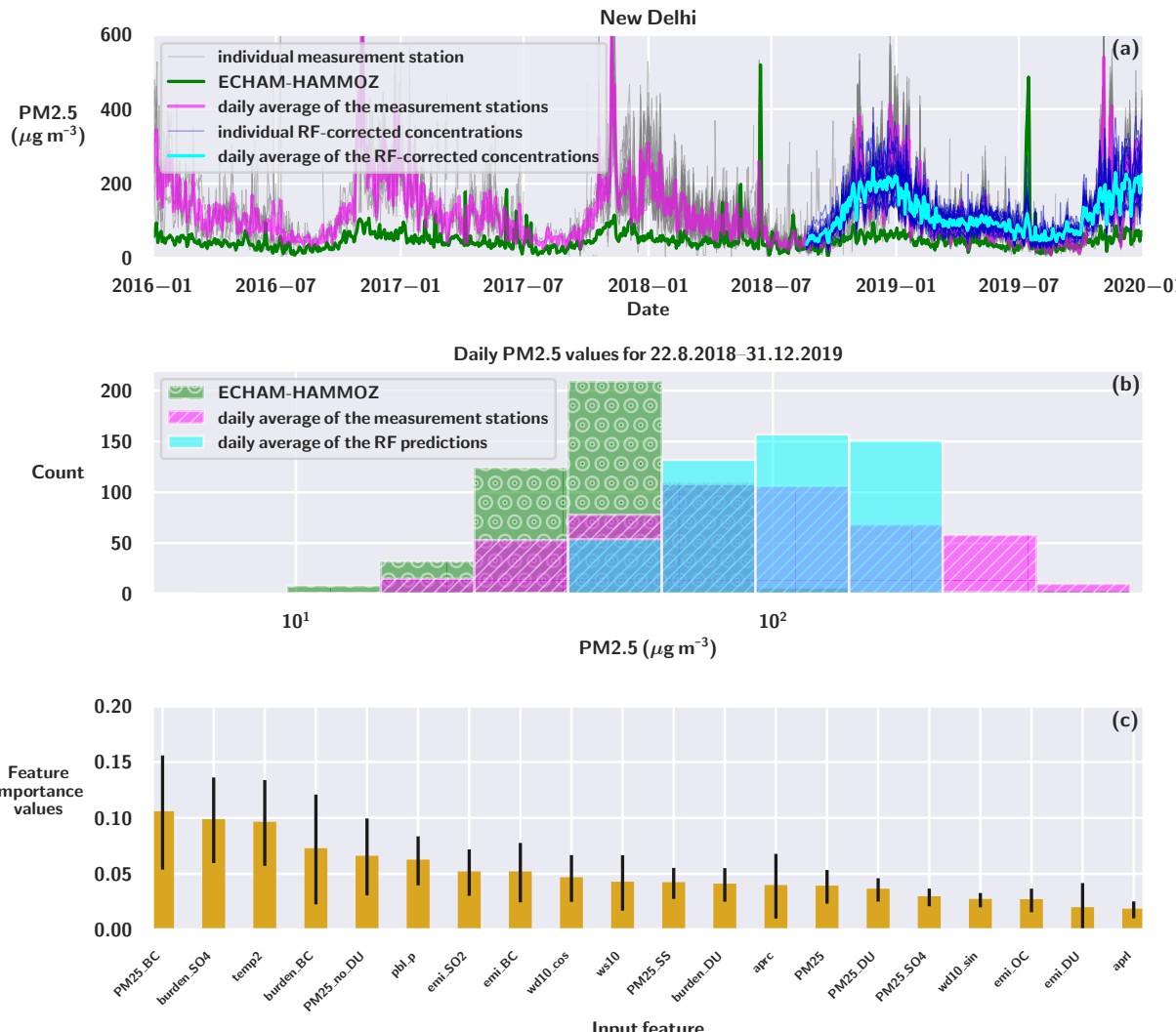

**Figure 1.** The results for random forest regression fitting for the RF_TRAIN simulation. (**a**) The solid green line represents ECHAM-HAMMOZ derived PM2.5 which includes mineral dust. The thinner grey lines represent individual measurement station data which were used for training. The pink line represents the daily average of all the stations. The thin dark blue lines show the individual RF-corrected PM2.5 values for each station, and the thicker turquoise line is the average of all the RF-corrected PM2.5. (**b**) The distribution of daily PM2.5 values for the testing phase (22.8-2018–31.12.2019). The green bars filled with white circles show the distribution of ECHAM-HAMMOZ daily PM2.5 concentrations. The pink bars with slashes indicate the daily average PM2.5 values for the average of the measurement stations. The turquoise bars show the daily average RF-corrected PM2.5 values. (**c**) The average normalized feature importance values for the input features are represented with yellow bars. The black whiskers show the standard deviation of individual RF model importance values.

As Fig. 1a shows, the random forest prediction captures long-term trends in the average measured PM2.5 concentrations very
320 well. The yearly cycle and seasonal variation are clearly improved when compared to the original ECHAM-HAMMOZ surface

PM2.5 concentration. The PM2.5 predicted by ECHAM-HAMMOZ was $(50.5 \pm 41.6)\,\mathrm{\mu g\,m^{-3}}$ for the testing period, where 50.5 indicates the mean and 41.6 the standard deviation. This was significantly lower than what was measured at the stations $((118.1 \pm 88.3)\,\mathrm{\mu g\,m^{-3}})$ for the same period. After the RF correction, the result $(118.8 \pm 52.0)\,\mathrm{\mu g\,m^{-3}}$ matched statistically very well with the measurements.

325  The biggest enhancement for the RF-corrected PM2.5 concentrations is for the late autumn (i.e. October and November) and winter (December and January) months, which are distinctly higher than the summer (June and July) values. These elevated wintertime PM2.5 values were much less pronounced in the uncorrected ECHAM-HAMMOZ PM2.5 concentrations.

  However, day-to-day variations do not correlate well with observations and extreme values in the RF-corrected data are much smaller in magnitude compared to measured extremes. This can be also seen by comparing the histograms presented in Fig. 330 1b. While the distribution for the average measured concentrations is relatively wide, the distribution for average RF-corrected model values is more narrow and centered around $100\,\mathrm{\mu g\,m^{-3}}$. This indicates that the RF-corrected PM2.5 is not ideal when the aim is to analyze very short-term air pollution episodes, but that the RF correction is more feasible for studying air pollution trends during longer time frames. Including more information about local conditions, like local orography and information of emission with higher spatial and temporal resolution might help to remedy these problems. Furthermore, Fig. 1a shows that 335 there is a fairly large variation between individual RF corrections, which are conducted separately for each station. There are large differences in measured PM2.5 concentrations between individual ground stations due to e.g. differences in local emission sources and micro climate. This heterogeneity in measured PM2.5 values translates directly into the RF corrections.

  We computed the feature importance values separately for each RF model used for individual ground measurement stations, and then computed the average value and standard deviation of the individual importance values. The average of feature 340 importance values for the RF models used in Fig. 1 are presented in Fig. 1c. The figure shows that the station-averaged feature importance values are relatively close to each other. The highest value is for PM2.5 due to BC $(0.105 \pm 0.051)$, but when the standard deviation is considered, the difference to some other features (e.g. 2 meter temperature or sulfate burden) is not distinct.

  Some of the input feature variables are correlated with each other. Even though we removed input features which were 345 correlated most strongly to other, included input features (see Section 2.6), the remaining correlated input features might affect the calculated values, especially considering the random component of RF. We decided to use all the input variables despite the correlation since this combination of input variables provided the best outcome of the RF fit. Furthermore, it might also depend on the station which input variables are more important for RF predictors. PM2.5 due to BC, for instance, results in highest or second highest feature importance value for many of the stations, while for some stations the value is less than 0.05. This may 350 also suggest that there may be differences in local meteorology and local emission sources between individual stations. Hence, forming an unique RF model for each station helps to maintain these local characteristics when conducting the downscaling procedure. Altogether, the importance values should be considered more as indicative variables.

  The error statistics for comparing the original ECHAM-HAMMOZ PM2.5 concentration to the average measured PM2.5 values, and for comparing the RF-corrected PM2.5 to average station PM2.5 values are presented in Table 4. The error mea-355 sures presented in Table 4 clearly show that the RF correction improves the PM2.5 representation for New Delhi. The Pearson

**Table 4.** The error statistics for the original ECHAM-HAMMOZ PM2.5 and the RF-corrected PM2.5. Root mean squared error (RMSE), mean relative error (MRE), mean absolute error (MAE), R-squared ($R^2$) and Pearson correlation.

| error measure | ECHAM-HAMMOZ PM2.5 | RF-corrected PM2.5 |
|---|---|---|
| RMSE ($\mu g\,m^{-3}$) | 112.75 | 56.48 |
| MRE (%) | −36.43 | 31.42 |
| MAE ($\mu g\,m^{-3}$) | 78.22 | 38.37 |
| $R^2$ | −0.63 | 0.59 |
| Pearson correlation | 0.19 | 0.80 |

correlation is almost 0.8 between RF-corrected PM2.5 concentrations and the average station PM2.5, whereas the same correlation for uncorrected ECHAM-HAMMOZ PM2.5 was 0.19. The root mean squared error (RMSE) for RF correction is almost half of the error for ECHAM-HAMMOZ, indicating that the occurrence of really high bias is smaller in the RF-corrected version. On the other hand, the mean relative error (MRE) for RF-corrected PM2.5 values is about the same magnitude as for original ECHAM-HAMMOZ PM2.5. This is mostly due to overestimation of the low summertime concentrations. As the yearly variation of PM2.5 concentrations in New Delhi is weak in ECHAM-HAMMOZ, the RF-corrected PM2.5 values also have a slightly dampened yearly cycle.

### 3.2 Applying RF to free-running simulations

The original PM2.5 values for simulations PRES, CLE_2030 and MITIG_2030 are presented in Fig. S2. The multi-year daily average values for the non-dust, uncorrected PM2.5 concentrations in ECHAM-HAMMOZ are shown in Fig. 2a. The results for RF-corrected PM2.5 values (i.e. data resulting from applying the RF bias correction to the ECHAM-HAMMOZ data) are presented in Fig. 2b. The corresponding feature importance values for the RF models applied are presented in Fig. 2c.

**New Delhi**

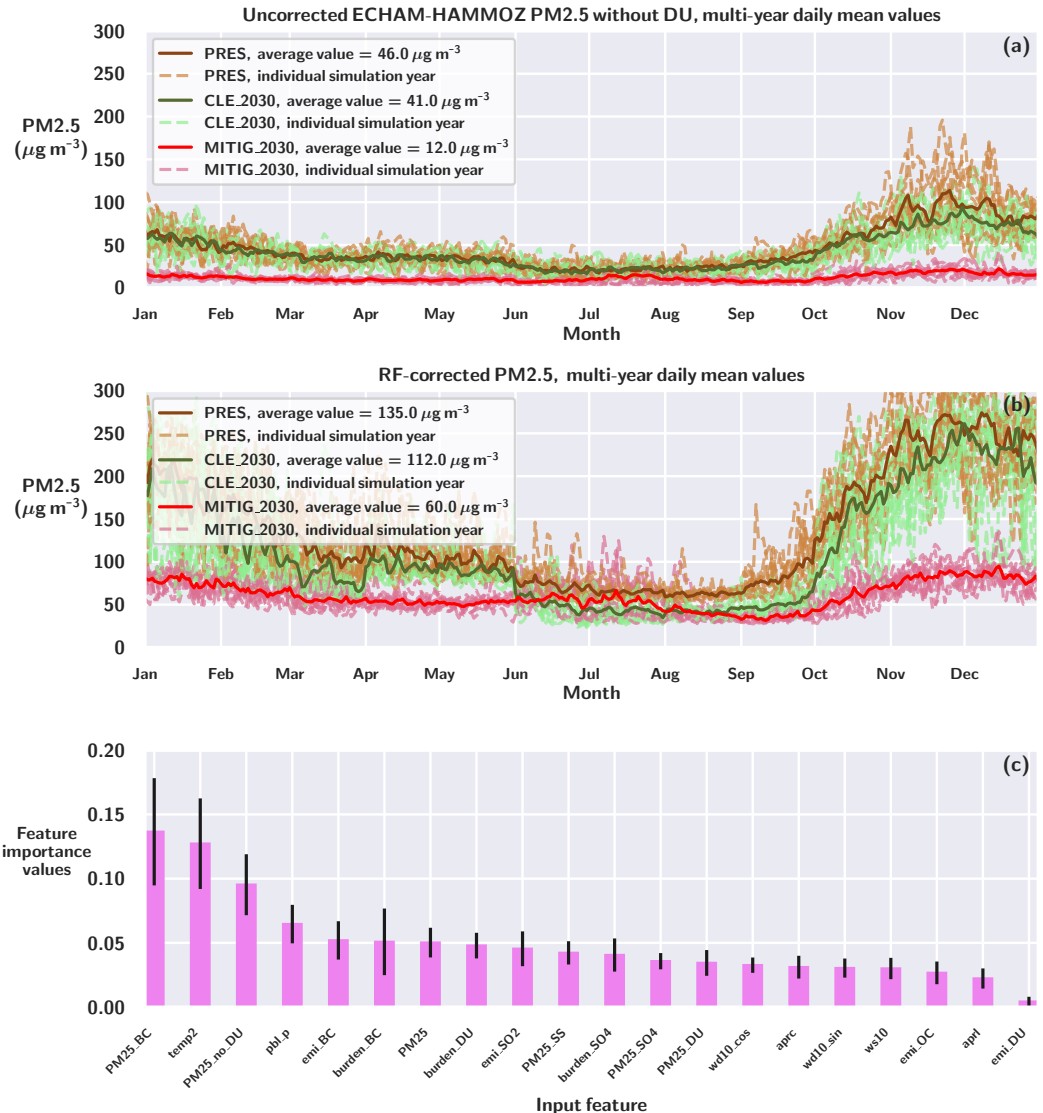

**Figure 2.** (**a**) The uncorrected ECHAM-HAMMOZ PM2.5 concentrations (excluding dust) for PRES, CLE_2030 and MITIG_2030 simulations as multi-year daily average values for each scenario. Individual years are marked with dashed curves, and the multi-year daily average values are shown with solid, colored lines. (**b**) The RF-corrected concentrations for PRES, CLE_2030 and MITIG_2030 simulations as a multi-year daily average PM2.5 values for each scenario. The dashed lines represent the prediction for each individual year, and the thick solid lines are the multi-year daily averages. (**c**) The average normalized feature importance values for the input features are represented with pink bars. The black whiskers show the standard deviation of individual RF model importance values.

The ECHAM-HAMMOZ non-dust PM2.5 for CLE_2030 is slightly smaller than the PRES non-dust PM2.5. This is most likely due to reductions in sulfur emissions in the New Delhi surroundings (see Table 1), although they are partly compensated by a small increase in the anthropogenic OC emissions. In the MITIG_2030 simulation, the OC emissions are much smaller than in the PRES simulation, which translates into a smaller local PM2.5 concentration. This shows that the OC particle mass contributes substantially to the total PM2.5 in ECHAM-HAMMOZ for the New Delhi region, and the large changes in OC influence the New Delhi PM2.5. Table 1 shows that the total OC emission mass is more than double the mass of emitted BC, both on a global and regional scale. Furthermore, the contribution of sulfate to the long-term PM2.5 in New Delhi is known be lower than the contribution of OC (Lalchandani et al., 2022), though sulfate might have a more significant role during winter haze periods.

As described in Section 2.2, the largest reductions for anthropogenic aerosol emissions under the future mitigation scenario MITIG_2030 were assumed to come from the domestic (BC & OC), waste (OC), traffic (BC), industry ($SO_2$) and energy ($SO_2$) sectors. The emissions from different sectors contribute differently to the atmospheric aerosol concentrations, because the assumed size distributions, mixing states and injection height at emission time are different. Each sector therefore also contributes differently to the modelled PM2.5 values.

Figure 2b presents the multi-year daily mean values of the RF-corrected PM2.5 concentrations for all of the simulations. For simulation CLE_2030 the values are slightly smaller than for PRES ($(112 \pm 73)\,\mathrm{g\,m^{-3}}$ vs. $(135 \pm 73)\,\mathrm{g\,m^{-3}}$). Moreover, the relative difference between the average RF-corrected concentrations for CLE_2030 and PRES is of about the same magnitude as for the original ECHAM-HAMMOZ concentrations in Fig. 2a. The average PM2.5 of the RF-corrected CLE_2030 is $17\,\%$ smaller than the average PM2.5 of the RF-corrected PRES. For the uncorrected ECHAM-HAMMOZ concentrations, CLE_2030 PM2.5 was $\sim 8\,\%$ smaller than the average value of the PRES PM2.5.

Figure 2b shows that stringent aerosol mitigation affects particularly wintertime (i.e. December and January) surface PM2.5 concentrations, which is why the RF-corrected PM2.5 values for simulation MITIG_2030 are much lower than for PRES or CLE_2030. The overall average PM2.5 concentration for RF-corrected MITIG_2030 is $(60 \pm 19)\,\mathrm{g\,m^{-3}}$, and the wintertime PM2.5 concentrations for MITIG_2030 are less than half of the corresponding values for CLE_2030 and PRES.

As seen in Fig. 1, with the given training data, the RF correction is not optimal for predicting day to day variations, resulting in smaller short-term variability than what the measurements show. This most likely also translates into the RF correction of the free-running simulations discussed in this section. Especially the RF-corrected MITIG_2030 multi-year average has little short-term variation throughout the year. This may be partly explained by the lower temporal resolution of the wildfire emissions in the free-running simulations, which were monthly average climatological values.

The RF-corrected MITIG_2030 PM2.5 for June and July partly exceed the CLE_2030 PM2.5 values, while the RF-corrected values obtained for the winter months (December and January) in CLE_2030 and PRES clearly exceed MITIG_2030. The elevated summertime MITIG_2030 PM2.5 is partly due to the additive bias correction approach, where the RF model adds a negative or positive bias term to the non-dust PM2.5 concentration. We can see from Fig. 2a that there is a small increase in the uncorrected non-dust MITIG_2030 PM2.5 for July. This was related to a minor increase in the summertime sea salt

concentrations in MITIG_2030, which was a result of increased wind speeds south–west from New Delhi (see Section 3.3). We assume that the RF model slightly magnifies this increase.

Moreover, we compared the average regional convective precipitation values for the New Delhi surroundings (see Section 2.6). The average convective precipitation for June and July was $(4.4 \pm 4.1) \times 10^{-5}\,\mathrm{mm\,s^{-1}}$ and $(4.2 \pm 4.1) \times 10^{-5}\,\mathrm{mm\,s^{-1}}$ for PRES and CLE_2030, respectively. For MITIG_2030, the average summertime convective precipitation was $(3.1 \pm 3.8) \times 10^{-5}\,\mathrm{mm\,s^{-1}}$. Based on the Mann-Whitney U-test, there was a statistically significant difference between MITIG_2030 and PRES summertime precipitation in the New Delhi surroundings. Decreased summertime convective precipitation indicates less washout, allowing for higher surface PM2.5 concentrations at the MITIG_2030 simulation.

Furthermore, the RF-corrected CLE_2030 summertime PM2.5 is distinctly lower than the corresponding PRES PM2.5. To investigate this difference, we further tested the effects of input variables on the RF correction by excluding individual input features from the training phase (not shown). The results showed that when the BC emissions were excluded from the RF model, the RF-corrected PM2.5 summertime values were at the same level for CLE_2030 and PRES, and the MITIG_2030 concentration was larger than with the default correction. This indicates that the BC emissions may have an effect in the RF model, although the BC emissions' feature importance values presented in 1c and 2c are not the largest values. The feature importance values indicate the contribution of a feature to the total reduction in the error criterion. However, importance values do not reveal the sensitivity of the RF model to specific input features as the reduction in error does not indicate directly how much a feature affects the RF model output trends and magnitude. All in all, the RF model prediction is not a strong function of any of the individual features, but a complex and a non-linear output of multiple regression trees. That is why the trends in the RF-corrected PM2.5 values partially differ from the uncorrected non-dust PM2.5 values.

In order to analyze how the seasonal trends are captured in the RF-corrected PM2.5 values, we calculated ratios between winter (December & January) and summer (June & July) averages. For the RF-corrected PRES concentrations, the ratio was 3.2, which is very close to the ratio for measurement stations ($\sim 3.2$). The same ratio for the uncorrected PRES PM2.5 concentrations was only 1.7, which indicates that the RF correction clearly improved the difference between the seasonal values. The ratio for RF-corrected CLE_2030 was 4.3, due to the lower summertime concentrations compared to PRES. For RF-corrected MITIG_2030, the ratio was 1.5, which is less than half of the ratio of the measurements.

The feature importance values presented in Fig. 2c show that local temperature and the BC component of PM2.5 are important factors for the applied RF model. In contrast to the normalized importance values presented in Section 3.1, the sulfate burden is not as strong a predictor feature for this training set combination. This indicates that the choice of time period for input data can affect the RF correction up to some extent. Moreover, none of the input features had an average importance value over 0.13, which suggests that the algorithm is able to utilize information from most of the variables. Additionally, as mentioned in Section 3.1, the correlation between individual input features might have an effect on importance value calculations.

## 3.3 Radiative effects

We calculated the 2D $\mathrm{RF_{ARI}}$ and ERF values in order to visualize how changes in anthropogenic aerosol emissions affect the regional radiative balance at the top of the atmosphere. The radiative forcing values are only due to changes in anthropogenic

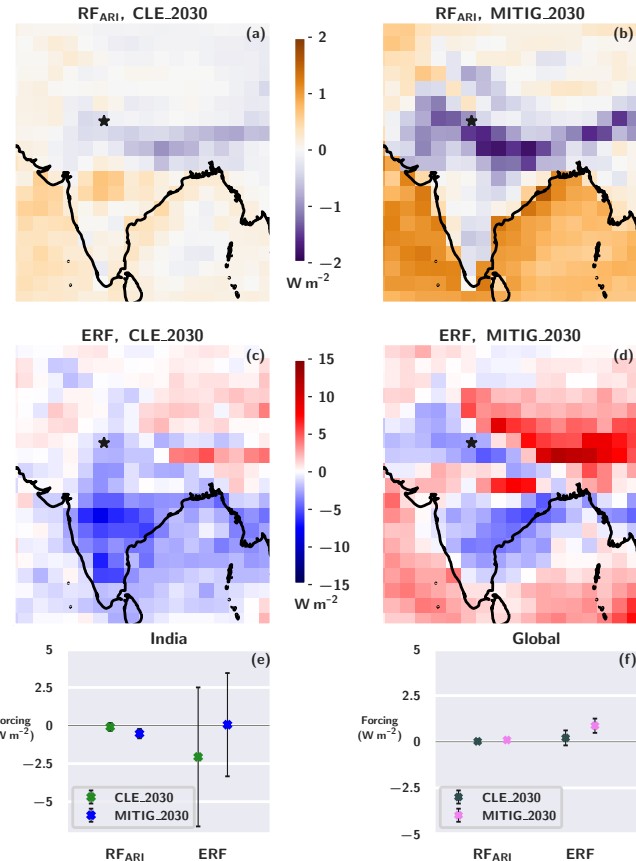

**Figure 3.** The spatial $RF_{ARI}$ and ERF for the CLE_2030 (**a,c**) and MITIG_2030 (**b,d**) simulations. The Indian (**e**) and global averages (**f**) for $RF_{ARI}$ and ERF are shown with green, blue, dark grey and pink markers, respectively. The whiskers represent the combined standard deviation.

aerosol emissions, as all other anthropogenic climate forcers (e.g. emissions, GHG concentrations) were kept constant. The $RF_{ARI}$ and ERF for CLE_2030 and MITIG_2030 simulations were calculated as presented in Section 2.8, using the uncorrected ECHAM-HAMMOZ outputs for radiative fluxes. Both regional 2D values and the area-weighted mean values for India and for the whole globe are presented in Figure 3.

The Indian $RF_{ARI}$ values for CLE_2030 in Fig. 3a are fairly small, but show a clear regional distribution with mostly negative values in the northern part of India and mostly positive values in the central and southern parts of India. The $RF_{ARI}$ values over the ocean regions surrounding India are close to zero or slightly positive. In MITIG_2030, on the other hand, the $RF_{ARI}$ values over India are mostly negative with values close to the Himalaya exceeding $-2\,\mathrm{W\,m^{-2}}$. This is in sharp contrast to the ocean regions surrounding India, where the $RF_{ARI}$ values are positive with extremes exceeding $+2\,\mathrm{W\,m^{-2}}$.

Altogether, the area-weighted average $RF_{ARI}$ values in Fig. 3e show that the $RF_{ARI}$ for India is negative for both CLE_2030

$((-0.09\pm0.26)\,\mathrm{W\,m^{-2}})$ and MITIG_2030 $((-0.53\pm0.31)\,\mathrm{W\,m^{-2}})$. This result is somewhat surprising as both of the scenarios, CLE_2030 and MITIG_2030, included reductions in both absorbing and scattering aerosols. The net negative $\mathrm{RF_{ARI}}$ for India results from the complex interplay of the differing optical properties of the different aerosol compounds as well as the surface below. On the one hand, anthropogenic BC emissions in MITIG_2030 are much lower than in CLE_2030, making a

cooling contribution to $\mathrm{RF_{ARI}}$ (less absorption of SW radiation). On the other hand sulfur and OC emissions are reduced as well, which makes a warming contribution to $\mathrm{RF_{ARI}}$ (less scattering of SW radiation). In addition, at the top of the atmosphere (TOA), the cooling effect of BC reduction is enhanced over brighter backgrounds (e.g. clouds and snow), while the warming effect of OC and sulfur reductions is enhanced over darker backgrounds (e.g. ocean and forests). This explains the sharp contrast in Fig. 3b — over the dark ocean, the warming contribution of OC and sulfate reductions dominates, leading to positive

$\mathrm{RF_{ARI}}$ values. This effect may be enhanced due to sulfur reductions in the shipping sector.

Over the Indian land areas, surface albedos are on average much brighter, which makes the cooling effect of BC reductions dominate. Over northern India, at the foot of the Himalayas, $\mathrm{RF_{ARI}}$ in MITIG_2030 is especially negative. This might be explained with the overall very high aerosol concentrations in this region (Murari et al., 2015; Jethva et al., 2019; Bera et al., 2021), which frequently leads to thick haze episodes (Saikawa et al., 2019). Strong haze acts like a bright background (e.g.,

a low cloud), thereby enhancing the cooling effect of BC emission reductions. While OC contributes strongest to the surface PM2.5 loadings (see Section 3.2), BC makes the strongest per unit mass contribution to the radiative forcing and hence can outweigh the cooling effects of OC and SO4, especially over bright backgrounds. The $\mathrm{RF_{ARI}}$ is more negative for the MITIG_2030 simulation, even though the haze layer due to anthropogenic aerosols is stronger in CLE_2030. This is because BC is reduced more heavily in MITIG_2030, which leads to stronger contributions to $\mathrm{RF_{ARI}}$ than the changes in the background aerosol

layer.

Furthermore, due to interaction with radiation, anthropogenic aerosols can modify regional weather patterns. As Wei et al. (2022) pointed out, the decreasing BC-containing aerosols from northern India result in easterly wind anomalies, which further diminish dust transportation from Thar desert, for instance. In our ECHAM-HAMMOZ simulations, we found an opposite effect when comparing MITIG_2030 and PRES (not shown), which can be due to simultaneous reduction of both scattering and

absorbing aerosol. The 10 meter wind speeds were on average slightly stronger south–west of New Delhi for the MITIG_2030 simulation, and the direction was shifted more towards the east. Since the surface wind speeds are the main inputs for the online dust model in ECHAM-HAMMOZ, the dust emissions were elevated for MITIG_2030 compared to PRES. This causes the dust burden over North–West India to be moderately larger in MITIG_2030 than in PRES. The dust aerosol in ECHAM-HAMMOZ are mostly scattering SW radiation, and the absorption effect due to mineral dust has a minor role. Therefore, the

increased DU burden in MITIG_2030 could potentially explain a small part of the negative $\mathrm{RF_{ARI}}$. However, this result is highly uncertain due to model internal variability of dust emissions and wind patterns, and analysing this phenomena further would require a much longer climate model simulation. Das et al. (2020) reported an increase in dust burden over central India, due to decreases in summertime precipitation. Our results also showed a decrease in convective precipitation when comparing MITIG_2030 values to PRES (see Section 3.2), which may have contributed to the increased dust loadings in MITIG_2030.

However, the modelled natural variation in precipitation patterns is large, and therefore there is significant uncertainty related to this mechanism.

The ERF values over India in Figs. 3c and d are mainly negative, contrary to what typically is expected as a net effect for aerosol reductions. The strong negative $RF_{ARI}$ for the MITIG_2030 simulation can be seen also in the ERF values, emphasizing how mitigating BC-rich sources is an efficient way to reduce TOA radiative forcing over India. However, the ERF signal is not spatially homogeneous over India, but there are differences between e.g. North–East and North–West. Furthermore, as ERF includes, in addition to $RF_{ARI}$, also aerosol-cloud interactions and rapid adjustments, the year-to-year variation is considerable and hence the area-weighted ERF values from Fig. 3e have large uncertainty ranges. The average ERF for India is $(-2.1 \pm 4.6)\,\mathrm{W\,m^{-2}}$ for CLE_2030 and $(0.06 \pm 3.39)\,\mathrm{W\,m^{-2}}$ for MITIG_2030, and is more negative for CLE_2030 than for MITIG_2030. This difference is a sum of complex mechanisms: on the one hand, compared to PRES, the OC and $SO_2$ emissions are distinctly lower in MITIG_2030 whereas in CLE_2030 the OC emissions remain almost the same and the $SO_2$ emissions are reduced by only about $25\,\%$. This means that, compared to CLE_2030, there is less scattering aerosol in MITIG_2030, to reflect SW radiation back to space. On the other hand, there is also less absorbing BC in MITIG_2030 compared to CLE_2030. There is also variability in dust emissions, which affects the ERF values. In addition, the changing aerosol concentrations also affect $RF_{ACI}$ and rapid adjustments.

We analyzed the changes in SW absorption between different simulations, and the results are presented in Fig. S3. Figure S3a shows that there is a small decrease in atmospheric absorption of SW radiation when comparing CLE_2030 and PRES. For MITIG_2030, there is clear signal of decrease in absorption as Fig. S3b shows. The areas for which we observed the largest changes in atmospheric SW absorption are almost the same regions as obtained by Prakash et al. (2020), even though their analysis handled only the BC emissions due to road transport. The aerosol emission reductions lead to larger surface forcing values, shown in Figs. S3c and d, as more SW radiation reaches the surface. The net TOA SW forcing in Figs. S3e and f describes the total amount of SW radiation that is absorbed either by the atmosphere or the surface layer. Over India, most of the SW portion of ERF is negative, which indicates that the net absorbed radiation is less in the CLE_2030 and MITIG_2030 simulations compared to the reference simulation PRES. Permadi et al. (2018) conducted a similar study for BC emissions reductions in Indonesia and Thailand. Their outcome was that along improved air quality, emissions reductions led to a decrease in absorption of SW radiation by BC, and therefore the resulting radiative forcing was negative.

The changes in aerosol emissions affect local cloud cover, which in turn has an impact on the radiative balance. One result of aerosol mitigation is a reduced Twomey effect (Twomey, 1977), i.e. when there are less aerosol particles to act as cloud condensation nuclei, at a theoretically constant liquid water path, clouds have less cloud droplets and therefore scatter less of the incoming SW radiation. When comparing CLE_2030 and PRES, the changes in aerosol concentrations translate surprisingly into a very small increase in the cloud droplet number concentration (CDNC) burden over most of India (not shown), i.e. the mean CDNC burden values are higher in CLE_2030 than in PRES for large areas over India. This would indicate that there are more cloud droplets to scatter SW radiation. This can potentially explain a small part of the negative ERF for CLE_2030. For the MITIG_2030 simulation, the CDNC burden over India decreases when compared to PRES simulation, which could partly

explain why in some areas in India the ERF values are positive. A more detailed analysis of the ACI-contribution to the ERF is out of the scope of this article.

O'Connor et al. (2021) simulated the present-day aerosol forcing compared to pre-industrial times with the UKESM climate model. They obtained that the aerosol ERF was positive over India, dominated by the strong positive forcing caused by present-day BC absorption. Similarly, Ramachandran et al. (2022) analyzed from a subset of CMIP6 model simulations that multi-model $ERF_{ATM}$ (ERF within the atmosphere, i.e. the part of ERF that excludes surface absorption) is positive over India for the year 2014 compared to pre-industrial times. Our results underline these findings, as we obtained that future reductions in the BC emissions would lead to a negative radiative forcing for most of India when compared to present-day emissions. However, as Ramachandran et al. (2022) concluded, there is a discrepancy between climate models and measurements when it comes to magnitudes and trends in aerosol optical depth (AOD) and single scattering albedo over Asia. Wang et al. (2021) also suggest that the representation of Asian aerosol trends in CMIP6 models is incorrect between 2006 and 2014, This indicates that CMIP6 climate models overestimate the Asian aerosol concentrations for the present-day conditions. However, as mentioned in Whaley et al. (2022), the recent decline in emissions from Asia is taken into account in the ECLIPSE V6b emissions, which suggests the overestimation of present-day aerosol concentrations should not be a major concern in our simulations.

On a global scale, Fig. 3f shows that the global $RF_{ARI}$ for both CLE_2030 and MITIG_2030 is positive. However, the standard deviation for CLE_2030 $RF_{ARI}$ is larger than the actual signal. The global average $RF_{ARI}$ is larger for MITIG_2030 $((0.09 \pm 0.04)\,\mathrm{W\,m^{-2}})$ than for CLE_2030 $((0.02 \pm 0.03)\,\mathrm{W\,m^{-2}})$. The slightly positive global $RF_{ARI}$ values are most likely due to the strong reductions in sulfur emissions (see Table 1). However, the simultaneous changes in BC emissions counterbalance some of the sulfate and OC reductions, which is why the $RF_{ARI}$ values are smaller than they would be if there would be only sulfur reduced. The same applies to the global ERF values. The overall global ERF for CLE_2030 is $(0.21 \pm 0.44)\,\mathrm{W\,m^{-2}}$, whereas for MITIG_2030 simulation, the global ERF is $(0.87 \pm 0.41)\,\mathrm{W\,m^{-2}}$.

Even though our future aerosol emission scenarios included simultaneous mitigation of BC, OC and $SO_2$, the GHG concentrations were assumed to remain at present-day levels. Therefore, the ERF and $RF_{ARI}$ values calculated for CLE_2030 and MITIG_2030 do not consider expected changes in GHGs, which would have potentially shifted the radiative forcing values towards less-positive values. For instance, Smith and Mizrahi (2013) estimated the maximum methane reductions would lead to a global radiative forcing of approximately $-0.08\,\mathrm{W\,m^{-2}}$ by 2030. Similarly, Smith et al. (2020) concluded that maximum feasible reductions in methane emissions could bring a $-0.23\,\mathrm{W\,m^{-2}}$ global radiative forcing in 2040 compared to a reference scenario without additional climate policies.

Furthermore, our climate modelling cases had only ten years of simulated data, which is a relatively short period for computing the ERF values for simulations with freely evolving wind fields. Hence, the modelled climate variability between distinct simulation years translates into wide uncertainty ranges for both global and regional ERF. All in all, the $RF_{ARI}$ and ERF values in Fig. 3 highlight that reducing anthropogenic aerosol emissions can lead to spatially heterogeneous forcing signals, depending on the region and the local conditions. The global ERF for MITIG_2030 was positive, whereas for the Indian territory the ERF and $RF_{ARI}$ values were mostly negative or close to zero.

## 4 Summary and Conclusions

In this study, we used a machine learning method to downscale fine particle (PM2.5) concentrations from the global scale aerosol-climate model ECHAM-HAMMOZ for surface PM2.5 values in the Indian mega-city New Delhi. We applied random forest (RF) regression for downscaling, and used measured PM2.5 values from various ground stations located in New Delhi for training the RF model.

Like many other global-scale climate models, ECHAM-HAMMOZ underestimates surface PM2.5 concentrations for several urban regions. There are various factors in global-scale modelling that lead to this, coarse horizontal and vertical resolution being one of the dominant reasons. Global climate models are by design intended to represent average concentrations over larger areas than what urban cities typically occupy. This is why their output values often deviate from measured point concentrations from urban measurement stations. However, global climate models are good at modelling long-term climate effects of air pollutants, which are important to consider when evaluating the changes in pollutant levels and improvements in air quality. With the help of statistical downscaling, we were able to employ ECHAM-HAMMOZ for analyzing the effects of aerosol emission mitigation on local air quality, and also estimate the corresponding regional and global radiative forcing values that affect future global warming progression.

The RF downscaling clearly improved the comparison between measured and modelled surface PM2.5. Especially the yearly cycle and seasonal differences are much better captured in the RF-corrected PM2.5 concentrations when comparing to the uncorrected ECHAM-HAMMOZ PM2.5 values. In addition, the overall correlation between RF-corrected and measured PM2.5 concentrations was considerably higher than between the uncorrected PM2.5 from ECHAM-HAMMOZ and measured values. However, the RF-corrected concentration values contained less extreme short-term variation (e.g. daily low and high values) compared to the ground station data. The very high values (i.e. pollution episodes) are important when analysing the health effects due to air pollution, since in addition to long-term exposure, high concentration short-term exposure is also known to bring negative health effects (Wei et al., 2019). This indicates that, with the training data used here, our RF downscaling correction is very well suited for applications which aim to analyze long-term trends in air pollution.

We further applied the RF model to free-running climate model simulations that had different scenarios for anthropogenic aerosol emissions, namely black carbon (BC), organic carbon (OC) and sulfur dioxide ($SO_2$). The aim was to analyze simultaneously both local air quality in New Delhi and regional radiative forcing under two future projections, business-as-usual 2030 (CLE_2030) and stringent mitigation in 2030 (MITIG_2030) scenarios, and compare those to the present-day simulation (PRES). The PM2.5 concentrations for New Delhi were of the same order of magnitude for CLE_2030 and PRES, both before and after the RF correction. Although the sulfur and BC emissions were smaller in CLE_2030, the OC emissions remained almost constant between CLE_2030 and PRES. A large part of PM2.5 mass is due to OC, which is why the difference between CLE_2030 and PRES was small. The PM2.5 concentrations for MITIG_2030 were remarkably lower than for PRES, which was to be expected, as all the anthropogenic aerosol emissions were mitigated strongly. The results indicated that the RF downscaling is a solid method for fixing resolution biases in local surface concentrations from global-scale modelling data.

However, the downscaled results showed reduced short-term variation, which causes a slight overestimation for some of the summertime concentrations and underestimation of some of the highest wintertime values for the New Delhi region.

In addition, we computed the corresponding regional and global radiative forcing values for the two future aerosol emission simulations. Over India, the forcing due to aerosol-radiation interactions ($RF_{ARI}$) was negative over northern India for both CLE_2030 and MITIG_2030. This was due to the dominating cooling contribution of the large reductions in BC emissions, even though the scattering aerosol compounds, OC and sulfate, were decreased as well. In addition, the effective radiative forcing (ERF) values were mostly negative over India for both CLE_2030 and MITIG_2030.

Globally, the average radiative forcing values were positive for both the CLE_2030 and MITIG_2030 simulations. However, the net radiative forcing values were spatially heterogeneous due to differences in emission levels and local environmental conditions, as the obtained results for India show. Our results emphasize that, opposite to the effects of reducing well-mixed GHGs, the impacts of aerosol mitigation can differ substantially between regions. For most of India, reducing aerosol emissions resulted in negative forcing values, which suggests a net cooling effect due to aerosol mitigation. This suggests that, along with air quality improvements, aerosol mitigation could potentially bring co-benefits to slowing down regional climate warming over large areas in India.

The downscaling method applied in this study employed only one type of machine learning algorithm. Further work with the downscaling model could combine multiple machine learning and regression algorithms as an ensemble model (Di et al., 2019). Likewise, our RF model could be developed towards a cascade-based RF (Yang et al., 2020), which might reveal some higher order correlations from the input training data. Furthermore, the method we proposed could be further advanced by including very detailed information about local infrastructure and traffic statistics, for instance. This could potentially improve the correction of short-term trends in New Delhi PM2.5. As a continuation to this study, one could apply the downscaling method described here during the ECHAM-HAMMOZ simulation (instead of after, as it was done here) and for larger areas. With some extensions which address additional aspects like, e.g., information about aerosol size, this may allow the bias correction to also affect the computation of other aerosol impacts, like radiative fluxes. However, such a dynamical approach for larger areas would also require a much larger spatial coverage of observational data, which would make the model computationally more demanding. Furthermore, without proper evaluation, such a model extension might introduce further uncertainties to the radiative forcing estimation.

Correcting local air pollution levels from a global-scale model has two main advantages. First of all, one can conduct simulations with global coverage and for long time periods, which may lower the total computational costs compared to modeling with high resolution regional scale models. With the help of downscaling, the global model data can be then rectified with the help of local ground station data. Secondly, this type of approach enables analyzing the effects of aerosol emission mitigation on both local air quality and Earth's energy budget simultaneously. Ideally, this improves the applicability of global climate models on research questions concerned with multiple effects of local aerosol emission reductions. As there is an increasing number of countries that have included BC in the Paris Agreement National Determined Contributions, the relevance of addressing both local and global effects of country-level mitigation of BC and co-emitted aerosol species is even greater. Furthermore, downscaling global model products instead of using finer resolution models might require less computational

capacity, and thereby decrease the energy consumption of modelling. This in turn could help to reduce the carbon footprint of climate modeling, which is a critical aspect in the era of decarbonizing anthropogenic activities (Lannelongue et al., 2021; Stevens et al., 2020).

To summarize, we obtained greatly improved seasonal trends for surface PM2.5 when applying machine learning downscaling to ECHAM-HAMMOZ data for New Delhi. Stringent, global aerosol mitigation resulted in improved air quality in New Delhi and negative radiative forcing for most of India. Organic carbon emissions had a stronger influence on local air quality, whereas black carbon emissions contributed more to the radiative effects.

*Code and data availability.* The ECHAM6-HAMMOZ model is made available to the scientific community under the HAMMOZ Software License Agreement, which defines the conditions under which the model can be used. The license can be retrieved from https://redmine.hammoz.ethz.ch/attachments/291/License_ECHAM-HAMMOZ_June2012.pdf. The model data can be reproduced using ECHAM-HAMMOZ model revision 6588 from the repository https://redmine.hammoz.ethz.ch/projects/hammoz (HAMMOZ consortium, 2019). The settings for the simulations and the Python scripts for data-analysis are given in the Fairdata Etsin service https://etsin.fairdata.fi/dataset/1968fb7e-4eb4-4f69-922b-acb9a33081ed . All emission input files, except ECLIPSE V6b are ECHAM-HAMMOZ standard and are available from the HAMMOZ repository (see https://redmine.hammoz.ethz.ch/projects/hammoz, HAMMOZ consortium, 2019).

ECLIPSE V6b data files are available at https://previous.iiasa.ac.at/web/home/research/researchPrograms/air/ECLIPSEv6b.html (last access: on the 8th February, 2023). The ground measurement station data for New Delhi can be downloaded from https://app.cpcbccr.com/ccr/#/caaqm-dashboard/caaqm-landing (last access: on the 8th February, 2023).

*Author contributions.* TM performed all the simulations. TM, AL, HK, KEJL and TK designed and planned the study. TM, AL, TK and HK conducted the data analysis and formulated the RF modeling setup. A-PH and VKS provided ground measurement data for the New Delhi region. All authors contributed to the writing process.

*Competing interests.* The authors declare that they have no conflict of interest.

*Acknowledgements.* This work was financially supported by the Academy of Finland Center of Excellence programme (grant no. 307331), Academy of Finland Flagship (grant no. 337552), Academy of Finland projects no. 335562, 308292 and 283031, European Research Council (ERC Starting Grant 678889), the Vilho, Yrjö and Kalle Väisälä Fund and the University of Eastern Finland Doctoral School Program in Environmental Physics, Health and Biology. For computational resources, we acknowledge CSC – IT Center for Science, Finland. The ECHAM-HAMMOZ model is developed by a consortium composed of the ETH Zürich, Max Planck Institut für Meteorologie, Forschungszentrum Jülich, University of Oxford, Finnish Meteorological Institute, and Leibniz Institute for Tropospheric Research, and is managed by the Center

for Climate Systems Modeling (C2SM) at ETH Zürich.

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
