# Peer review of "Assessing the climate and air quality effects of future aerosol mitigation in India using a global climate model combined with statistical downscaling"

_Atmospheric Chemistry and Physics, 2022_

## Referee Comment (RC2)

**Review of "Assessing the climate and air quality effects of future aerosol mitigation in India using a global climate model combined with statistical downscaling"**

The manuscript explores the possibilities of using a global climate model to investigate the effects of aerosol mitigation in India. A machine learning (ML) approach using Random Forest regression is used to downscale PM2.5 concentrations over a polluted city, New Delhi with the help of measured PM2.5 concentrations. Different PM loading future scenarios are projected and compared with the uncorrected and ML-corrected model outputs. The effects of aerosol mitigation are investigated in terms of radiative effects and effective radiative forcing under the PM future scenarios. The authors claim the improvement of global-scale model output in simulating the PM2.5 concentration over a small domain and their effectiveness in estimating the radiative impacts. The study demonstrates the potential of the emerging technique of ML in improving the large-scale model output in the process of statistical downscaling. The study is relevant and unique as mentioned above, and has a significant contribution to the relevant scientific domain. However, some concerns remain significant and need to be considered before publishing.

**General Comments**

The manuscript focuses on two aspects. (1) Demonstration of an ML technique in improving a global-scale model to simulate the PM2.5 concentrations over a small region via statistical downscaling under different emission scenarios. (2) Estimating the radiative effects of future aerosol scenarios using the RF-corrected model simulations. The manuscript structure is difficult to follow until reaching the present 'Conclusions' section which is not a conclusion, but a nice overview/summary of the entire work. If the authors want to highlight their simulation results regarding the impact of future aerosol mitigation, more discussion is needed with proper references to the existing findings, else it remains as a technical paper demonstrating the potential of ML in statistical downscaling. Currently, the physical mechanisms for some of the simulation outcomes are not given/found, but some tentative reasons are proposed. Many studies are documented the current aerosol-impact scenario using multiple scientific techniques (insitu, remote sensing, etc.) and future projections also for the Indian region. To ascertain the second aspect of the current manuscript, the first part needs to be flawless and should be explained confidently. Many parts of the manuscript are confusing which calls for further explanations for the smooth reading. One of the highlights of the findings is that the improvement of air quality is mostly due to the reduction of OC loading. However, the negative radiative forcing is attributed to the reduction of BC emission. This is an example of confusion arising while going through the manuscript. The map of India shown in Fig. 3 is not matching with the maps published by the institutions that provided the insitu data nor with one of the authors' affiliated institution. Please correct the map as per the source or remove the political boundaries as per the journal's recommendations. Language also needs improvement. The main concerns are listed below.

Methodology:

- Why does section 2.2 stand apart from sections 2.6 and 2.7?
- The hyper parameters were adjusted using different combinations based on the best error statistics. Can you please show the performance of the validation test data?
- What do you mean by setting the depth of each tree to infinity? How can you make sure of avoiding over-fit while keeping the depth of the tree as infinity? What are the criteria for fixing the number of

trees? It is said that a default value of 100 is taken as the number of trees in the present study. Why 100?

- Why the feature importance values are normalized, by doing so what is the chance of smoothing the non-linearity of the dependence of the input variables? Isn't there any criteria to fix the number of input variables? As the authors have pointed out the input variables are mutually correlated which is obvious in the atmosphere, including all of them may lead to over-fitting. What is the authors' claim on this point?
- For the global-scale modelling, ECLIPSE V6b emission scenarios are used. How appropriate is this emission inventory for simulating PM over the Indian region or what are the criteria for selecting this inventory for this study? What is the contribution of this inventory to the high under-estimation of PM loading by the model over Delhi as shown in the manuscript?
- How the exclusion of the mineral dust component solves the issue related to the PM2.5 peaking? How authors can make sure that this exclusion won't affect the other simulation results?
- Coming to the radiative forcing calculations (section 2.8), how is the definition given to the radiative forcing is related to the conventional definitions found in the published literature? If there is any difference please highlight and justify those, else give supporting citations.
- How the effective radiative forcing is estimated?

**Other comments**

L1: This opening sentence is misleading. The study demonstrates the potential of the ML technique in downscaling a global-scale mode output..

L6: You mean the model output is better than the measured PM2.5 values?

L11: This is a highly impactful statement. Better to give caution to the reader by mentioning the associated large uncertainty as seen in Fig. 3(e).

L38-40: As per the sentence, the role of ACI in aerosol indirect effects is undermined, hence please modify the sentence. Also, please explain the 'local meteorological dynamics' with references.

L72-73: Cannot find in any of the given references that 'emissions from New Delhi' significantly contribute to the ATAL. Please clarify.

L94: HAM 'threats' the chemical compounds..?

L158: Why do OC emissions increase by 2030 in CLE scenario while all others show a reduction?

L267: What is '2D yearly mean value'?

L356: If that is the case, what is the significance of feature importance values?

L385: This ___ somewhat?

L395-398: Confusing. The aerosol loading in MITIG_2030 is supposed to be lower than the CLE_2030, then how RF in MITIG_2030 is more negative than CLE_2030 at the Himalayan foothills. Bright background due to strong haze is expected to be more in CLE_2030.

L434-436: cannot understand. Do you mean that the CDNC burden was more in CLE_2030 scenario?

L435: Expand CDNC in the manuscript.

Conclusions: This section can be renamed as summary and conclusions by adding the significant findings of the study as bullet points.

---

## Author Comment (AC1)

**Responses to the referee comments:**
**Assessing the climate and air quality effects of future aerosol mitigation in India using a global climate model combined with statistical downscaling**

*Miinalainen, T., Kokkola, H., Lipponen, A., Hyvärinen, A.-P., Soni, V. K., Lehtinen, K. E. J., and Kühn, T.: Assessing the climate and air quality effects of future aerosol mitigation in India using a global climate model combined with statistical downscaling, Atmos. Chem. Phys. Discuss. [preprint], https://doi.org/10.5194/acp-2022-513, in review, 2022.*

We thank Anonymous Referee #1 and Anonymous Referee #2 for the efforts made to review our paper and for the valuable comments. Our responses are written below each comment separately. The referee comments are marked with yellow color and italic, and the author replies are marked with gray color.

**Replies to the comments made by the Anonymous Referee #1:**

*This manuscript presents global model simulation results over India from ECHAM for air quality and radiative forcing under present and future emission scenarios (from GAINS) through 2030. For the region covering Delhi, the results were downscaled using random-forest corrections using multiple emission, met, and orography variables.*

*While the radiative forcing calculations from ECHAM is a known path, the use of the same to bias correct and estimate air quality for a city is new. The later methods have been used, but not for model resolutions at 1.9 degrees. Machine Learning (ML) approach is a new and emerging field and the benefits of using a global model for both air quality and climate applications cannot be overlooked. While the methodology is well explained for correcting the model results with biases from ML, the statistics also improved after the corrections, the gaps between the measured and model-corrected numbers is still significant.*

We thank Anonymous Referee #1 for the useful comments.

As the referee mentioned, the RF-corrected PM2.5 does not fully reproduce the station-averaged PM2.5 for the testing phase data, especially over short time scales. This is analyzed and discussed in Section 3.1.

However, if the aim is to capture the long-term trends correctly, the RF-corrected PM2.5 performs as adequately as the PM2.5 obtained from the dispersion model System for

Integrated modeLling of Atmospheric coMposition (SILAM). In the Table below, we show the comparison statistics between modelled and mean of the New Delhi ground stations for the testing phase period 22.8.2018-31.12.2019.

| | Compared to the mean of the measurement station PM25 for 22.8.2018-31.12.2019, daily average values | | |
|---|---|---|---|
| | ECHAM PM2.5 | SILAM PM2.5 | RF-corrected PM2.5 |
| RMSE (µg/m³) | 113 | 65 | 56 |
| MRE (%) | -36 | 27 | 33 |
| MAE (µg/m³) | 78 | 46 | 39 |
| R² | -0.63 | 0.46 | 0.59 |
| Pearson correlation | 0.19 | 0.79 | 0.79 |

The error statistics show that the RF-corrected PM2.5 correlates with the measured data at the same level as SILAM PM2.5 does. Furthermore, the mean absolute error (MAE) and root mean squared error (RMSE) are smaller for RF-corrected PM2.5 than for SILAM data. The mean relative error (MRE) is smaller for SILAM than for RF-corrected data, but the difference is less than 10 percent units. This comparison suggests that the performance of the downscaling approach is comparable to air quality models if the aim is to utilize data to applications where long-term trends in air quality are central.

*The scenario analysis for air quality primarily hinges on the reproductive capacity of the model and the only question that is not clearly answered is why extract air quality data from such a coarse model (when the problem is known that coarser models have hard time replicating high-density urban areas with very distinct emission characteristics)? Especially, since FMI and IMD (author organizations) are known to conduct chemical transport modeling for air quality at much better resolutions globally and in India.*

The referee is correct that the downscaled PM2.5 does not fully correspond outputs from high resolution models when comparing day-to-day variation. However, this was not the aim of this project, and we presume that the concept of our manuscript was misinterpreted by the referee.

The main idea of our study is to expand the possibilities to utilize global model data in additional applications such as local air quality analysis. As we describe in the manuscript text (line 522 and forward), one advantage of using global-scale models for analyzing the effects of aerosol mitigation is that one can simulate fairly long time periods (decades to even a century), and that the simulations typically cover the whole globe. With the help of downscaling, one can "zoom in" to a very specific location and analyze how the global or local scale emission mitigation affects. Up to our understanding, air quality models are computationally more expensive due to the high grid resolution, and therefore the simulated time periods are shorter than with GCMs, and the simulation times can be much longer.

Furthermore, another advantage of using downscaling enhanced PM2.5 from a GCM is that one can simultaneously analyze the effects of aerosol mitigation on various other climatic processes. In our case, we focused on the radiative forcing, but one could, for instance, analyze global precipitation patterns or low-level cloud formation at very distinct regions in the globe.

One plausible application of downscaling PM2.5 could be simply to use it as a "quick tool" to evaluate how a mitigation scheme affects climate or air quality in different parts of globe. The intention of this study is not to present a method which can be used to replace chemical transport models. Instead, the main idea is to provide a relatively light-weight tool that can be used to assess simultaneously the climate and air quality effects of future aerosol emission reduction measures.

Furthermore, many air quality models require simulated data from global climate models for modelling the future climate. Regional modelling with fine grid resolution requires input for boundary conditions for large scale atmospheric dynamics, and therefore a separate global model simulation is needed before the actual air quality model simulation can be performed. Our intention was to explore if we could use the global model data directly also to analyzing surface air quality, especially if there is no need to study the underlying physical mechanisms in a detailed way.

We will update the Introduction text to communicate better our motivation and the core concept of the study.

*Why use a city like Delhi with so many stations with 0% data available in the ML testing phase? Why not use a city in Europe or the US with good availability rates and good representation of the sources, to show that the model is capable of replication after the bias corrections?*
We assume this is a misunderstanding as the majority of the stations used in this study have a coverage of 80 to 90% for testing phase (see Table S1). To emphasize this point, we will add a couple of statements about data availability in Section 2.3.

*The one drawback of the manuscript is the selection of the case study city (Delhi) -- which has strong seasonal trend, strong diurnal trend, and distinct sources (for SO2, BC, and OC) over the months. A city(s) or region(s) with consistent emission loads would cut down some uncertainty in the model and corrections methods and then apply to regions like India and China.*

We chose New Delhi as a case study city as it is ideal for this kind of study. The PM2.5 concentrations are at a high level for most of the year in New Delhi. Furthermore, Delhi National Capital Territory (NCT) is densely populated and relatively large in surface area. In addition, the ground measurement station network in New Delhi is very extensive and there was a sufficient amount of data available for our study.

Furthermore, we respectfully disagree with the referee that the unique characteristics backgrounding New Delhi pollution profile would make Delhi unsuitable for this kind of analysis. On the contrary, as we mention right in the first sentence of the abstract text, we aimed to study the potential of the downscaling procedure. As the referee described, the New Delhi PM2.5 is not constant all year around but has a lot of seasonal variation and strong dependence on anthropogenic sources. This makes New Delhi a good target region to explore how well downscaling can capture local tendencies, such as short- and long-term trends in PM2.5.

Anthropogenic aerosol emissions are relatively large in India (~15% of the global BC emissions), and they are not projected to decrease at a same rate as global emissions (see manuscript Table 1). That is why we considered that India and New Delhi are interesting areas to study, as the aerosol mitigation is expected to bring clear benefits to the local air quality, but the net radiative forcing due to simultaneous mitigation of three species (BC, OC and SO2) was unclear. We will update both the Introduction text and the text in Section 2.3 to give readers a clear impression of why New Delhi was chosen as a study of interest.

Unfortunately, it seems that we have not reported clearly enough in our manuscript that our bias correction model is meant to be built separately for each city. Though technically possible, it would require a very large global training dataset representing a wide variety of conditions to construct an ML-based model that would be able to generalize from one region to another. In practice, this type of global dataset does not exist. The characteristics of PM2.5 concentrations trends and how they depend on prevailing atmospheric conditions and the assumed local emissions are unique for each city/region. We used the same city for training and testing the RF Model to ensure representative training data for the model. We will emphasize this better in Section 2.7 and add a description of why there needs to be a separate model for each city.

*Line 237-242 and 290: It is not clear if the emissions and other variables extracted and used are still at the ECHAM resolution or further downscaled to support a region of 30km x 30km over Delhi? (L290) is an important observation - When making the bias corrections, besides the model grid variables, are there any variables that are seggregating the Delhi area signatures for a better fit?*
We thank the referee for making us aware that this aspect was not clear enough in the manuscript text. The input feature values used in the RF bias correction are all from one specific ECHAM-HAMMOZ simulation, and the data were not further downscaled to represent higher resolution grid data. Downscaling from T63 to L290 would require a large amount of external fine resolution data and would not serve the purpose of lowering the computational costs.

Instead, we post processed the input feature data by extracting values from the original ECHAM-HAMMOZ data of T63 horizontal resolution. We produced input feature data either by extracting data from one single grid box ('point'), or by summing ('fldsum') or averaging

('fldmean') over an area surrounding New Delhi (72 – 83 'W, 24 - 34 'N). This means that some of the input features represent more the regional changes in the simulated atmosphere, whereas the point features show more the local changes.

In our modeling set up, it is assumed that the measurement data from ground stations incorporate information about the very detailed characteristics affecting New Delhi pollution levels, such as urban infrastructure or very specific emission sources. Including detailed external data as an input feature (e.g., info about public traffic routes) would have reduced the feasibility of our modeling approach since external data is easily subject to change and would need to be kept updated constantly, or be based on additional scenarios or assumptions. Then again, the absence of detailed, local information is most likely the reason why our modelling approach is not capturing short-term variations very well. Developing such downscaling bias correction method that could also include external data could be an opportunity for future studies.

We will clarify these topics in the manuscript text in Section 2.6 and describe better the selection of input feature variables. We will also add discussion about potential future studies including very detailed information about local factors affecting air pollution levels.

*The results and conclusions of the study in terms of AQ and climate benefits of reducing emissions are as expected. However, since Delhi is the most polluted area/city in the world with not only a complex mix of emission sources, but also a complex mix of political and instititutional setup to manage these emissions. While the manuscript presented % changes (benefits for air quality and RF), the discussion doesn't include any explanation on how these % emission reductions will be acheived in the Delhi area. It is understood that the emissions work comes from a different model (GAINS). Since the manuscript very specifically mentions and analyzes data for one city only, it would be appropriate to also discuss this space.*

We would like to highlight here that it is not explicitly evident that future aerosol mitigation would bring a net negative forcing over India. As many studies have shown (for instance, Allen et al.,2020), simultaneous reductions in both scattering and absorbing aerosols are expected to reduce the net cooling effect of aerosols on a global scale.

The aerosol mitigation scenarios used in this study were part of the ECLIPSE V6b emission scenarios, which were designed with the GAINS model. The underlying assumptions in ECLIPSE are idealized in the sense that emission reductions are based on assumed perfect achievement of currently valid legislation (CLE) or full implementation of all currently available technologies to reduce emissions (MFR). In this sense these scenarios could be seen as maximum possible emission reductions from the 2020 viewpoint. Analyzing the underlying political actions needed to achieve the proposed emission reductions is beyond the scope of this study. A more detailed description of the ECLIPSE scenarios can be found from e.g. Stohl et al. (2015), Belis et al. (2022) and Klimont et al. (2017).

However, we will add a brief summary in Section 2.4 about the most significant source sectors reduced in the ECLIPSE MFR scenario. In addition, we will extend the discussion part in Section 3.2. to briefly mention possible source sectors that might have been contributing to air quality improvements the most.

*While there is merit to a new methodology to be able to model AQ data along with the climate data, the manuscript lacks punch and I am afraid that these correction results will be hard to replicate in another setting.*

While we appreciate the referee's feedback, we respectfully disagree that the proposed method would not be applicable for analyzing the effects in another city or region if trained and applied with suitable data. We do, however, recognize that the performance of the bias correction may be different for each target city.

As mentioned above, there might be a misunderstanding that we would propose to apply the model trained with New Delhi data to correct air quality levels for a different city. In case this referee's comment is based on that kind of assumption, we will elucidate in the updated text that the designed RF models are city specific.

In addition to the already mentioned improvements, we will go through the whole manuscript text and highlight the core concepts to sharpen the key messages of our study.

\* \* \* \* \* \* \* \* \* \* \* \* \* \* \* \* \* \* \* \* \* \* \* \* \* \* \* \* \* \* \* \* \* \* \* \* \* \* \* \* \* \* \* \* \* \* \* \* \* \* \* \* \* \* \* \* \* \* \* \* \* \* \* \* \* \* \* \* \* \* \* \* \*

**Replies to the comments made by the #Anonymous referee 2:**

*Review of "Assessing the climate and air quality effects of future aerosol mitigation in India using a global climate model combined with statistical downscaling"*

*The manuscript explores the possibilities of using a global climate model to investigate the effects of aerosol mitigation in India. A machine learning (ML) approach using Random Forest regression is used to downscale PM2.5 concentrations over a polluted city, New Delhi with the help of measured PM2.5 concentrations. Different PM loading future scenarios are projected and compared with the uncorrected and ML-corrected model outputs. The effects of aerosol mitigation are investigated in terms of radiative effects and effective radiative forcing under the PM future scenarios. The authors claim the improvement of global-scale model output in simulating the PM2.5 concentration over a small domain and their effectiveness in estimating the radiative impacts. The study demonstrates the potential of the emerging technique of ML in improving the large-scale model output in the process of statistical downscaling. The study is relevant and unique as mentioned above, and has a significant contribution to the relevant*

*scientific domain. However, some concerns remain significant and need to be considered before publishing.*

*General Comments*

*The manuscript focuses on two aspects. (1) Demonstration of an ML technique in improving a global-scale model to simulate the PM2.5 concentrations over a small region via statistical downscaling under different emission scenarios. (2) Estimating the radiative effects of future aerosol scenarios using the RF-corrected model simulations. The manuscript structure is difficult to follow until reaching the present 'Conclusions' section which is not a conclusion, but a nice overview/summary of the entire work.*

We thank the referee for the insightful suggestions to improve the manuscript structure and content. We will re-organize some of the subsections in Section 2 and rename Section 4 as suggested.

*If the authors want to highlight their simulation results regarding the impact of future aerosol mitigation, more discussion is needed with proper references to the existing findings, else it remains as a technical paper demonstrating the potential of ML in statistical downscaling. Currently, the physical mechanisms for some of the simulation outcomes are not given/found, but some tentative reasons are proposed. Many studies are documented the current aerosol-impact scenario using multiple scientific techniques (in situ, remote sensing, etc.) and future projections also for the Indian region.*

We will add more discussion and clarification about the potential physical mechanisms underlying the effects of mitigation and will also insert citations to relevant studies in Subsections 3.2 and 3.3. Further information is also included below in our answers to the detailed comments.

*To ascertain the second aspect of the current manuscript, the first part needs to be flawless and should be explained confidently. Many parts of the manuscript are confusing which calls for further explanations for the smooth reading.*

We thank the referee for bringing up that some parts of the manuscript may be confusing to the reader. We will go through the entire manuscript and improve the general readability of the text.

*One of the highlights of the findings is that the improvement of air quality is mostly due to the reduction of OC loading. However, the negative radiative forcing is attributed to the reduction of BC emission. This is an example of confusion arising while going through the manuscript.*

This was an excellent comment and helped us to understand which parts of our manuscript are confusing. The reason why OC influences air quality more than BC, though BC has stronger effects on the radiative balance, is related to the optical properties of these aerosol species.

The atmospheric mass load of BC is relatively small compared to the mass load of OC. The radiative forcing per mass load of BC, on the other hand, is much bigger than for OC, as the optical properties differ. That is why in this study the emission reductions of OC influence PM2.5 concentrations more, while BC-reductions have a larger impact on radiative forcing. We will update the text in Sections 3.2. and 3.3 and clarify how different aerosol species might have very different effects on radiative balance and air quality.

*The map of India shown in Fig. 3 is not matching with the maps published by the institutions that provided the insitu data nor with one of the authors' affiliated institution. Please correct the map as per the source or remove the political boundaries as per the journal's recommendations.*

We thank the referee for pointing out this issue. We will remove the political boundaries from Figures 3 and S3.

*Language also needs improvement. The main concerns are listed below.*

*Methodology:*

*• Why does section 2.2 stand apart from sections 2.6 and 2.7?*

We will move Section 2.2 below Section 2.5 and update the numbering of the subsections.

*• The hyper parameters were adjusted using different combinations based on the best error statistics. Can you please show the performance of the validation test data?*

This statement in the manuscript was somewhat inaccurate as the hyper parameter selection seemed not to affect the model performance substantially. The error statistics for testing

different RF parameter combinations are presented in the Figure below.

| max_depth: None, Max_features = 7 | nr of trees | 20 | 50 | 100 | 200 | 300 |
|---|---|---|---|---|---|---|
| | RMSE (µg/m³) | 56.56 | 56.52 | 56.43 | 56.52 | 56.54 |
| | MRE (%) | 32.29 | 31.30 | 31.98 | 32.21 | 32.26 |
| | MAE (µg/m³) | 38.69 | 38.46 | 38.57 | 38.57 | 38.65 |
| | R² | 0.59 | 0.59 | 0.59 | 0.59 | 0.59 |
| | Pearson correlation | 0.79 | 0.79 | 0.80 | 0.79 | 0.79 |

| max_depth: None, Max_features = None | nr of trees | 20 | 50 | 100 | 200 | 300 |
|---|---|---|---|---|---|---|
| | RMSE (µg/m³) | 56.79 | 56.81 | 56.90 | 56.84 | 56.76 |
| | MRE (%) | 33.15 | 33.52 | 33.46 | 33.08 | 33.11 |
| | MAE (µg/m³) | 39.02 | 39.09 | 39.16 | 39.09 | 39.01 |
| | R² | 0.59 | 0.59 | 0.58 | 0.59 | 0.59 |
| | Pearson correlation | 0.79 | 0.79 | 0.79 | 0.79 | 0.79 |

| max_depth: None, Max_features = 15 | nr of trees | 20 | 50 | 100 | 200 | 300 |
|---|---|---|---|---|---|---|
| | RMSE (µg/m³) | 56.73 | 56.43 | 56.45 | 56.45 | 56.53 |
| | MRE (%) | 32.15 | 32.54 | 32.63 | 33.27 | 32.62 |
| | MAE (µg/m³) | 38.57 | 38.61 | 38.69 | 38.81 | 38.75 |
| | R² | 0.59 | 0.59 | 0.59 | 0.59 | 0.59 |
| | Pearson correlation | 0.79 | 0.79 | 0.79 | 0.79 | 0.79 |

| max_depth: 10, Max_features = 7 | nr of trees | 20 | 50 | 100 | 200 | 300 |
|---|---|---|---|---|---|---|
| | RMSE (µg/m³) | 56.78 | 56.78 | 56.59 | 56.54 | 56.51 |
| | MRE (%) | 31.72 | 32.39 | 32.58 | 31.78 | 31.91 |
| | MAE (µg/m³) | 38.70 | 38.77 | 38.77 | 38.56 | 38.57 |
| | R² | 0.59 | 0.59 | 0.59 | 0.59 | 0.59 |
| | Pearson correlation | 0.79 | 0.79 | 0.79 | 0.79 | 0.79 |

| max_depth: 5, Max_features = 7 | nr of trees | 20 | 50 | 100 | 200 | 300 |
|---|---|---|---|---|---|---|
| | RMSE (µg/m³) | 56.93 | 57.00 | 57.12 | 57.09 | 57.12 |
| | MRE (%) | 31.65 | 31.18 | 31.80 | 31.87 | 31.76 |
| | MAE (µg/m³) | 38.77 | 38.76 | 38.99 | 38.96 | 38.96 |
| | R² | 0.58 | 0.58 | 0.58 | 0.58 | 0.58 |
| | Pearson correlation | 0.79 | 0.79 | 0.79 | 0.79 | 0.79 |

| max_depth: 15, Max_features = None | nr of trees | 20 | 50 | 100 | 200 | 300 |
|---|---|---|---|---|---|---|
| | RMSE (µg/m³) | 56.84 | 56.83 | 56.85 | 56.79 | 56.68 |
| | MRE (%) | 34.39 | 33.11 | 33.62 | 33.26 | 33.19 |
| | MAE (µg/m³) | 39.33 | 39.09 | 39.20 | 39.06 | 39.02 |
| | R² | 0.59 | 0.59 | 0.59 | 0.59 | 0.59 |
| | Pearson correlation | 0.79 | 0.79 | 0.79 | 0.79 | 0.79 |

As the statistics show, our RF model setup is not overly sensitive to the selection of hyper parameters. Hastie et al. (2009, p. 590) reported a similar tendency. Based on their experiences, random forests require very little tuning.

Furthermore, our modeling approach had 31 RF models (one per measurement station), and we used the same RF hyperparameters in each of them, as we did not tune each station-specific RF model separately. We therefore decided to use the default, recommended values in the RF models. We will update the manuscript text to describe the selection of hyper parameters in a more transparent manner.

● *What do you mean by setting the depth of each tree to infinity? How can you make sure of avoiding over-fit while keeping the depth of the tree as infinity? What are the criteria for fixing the number of trees? It is said that a default value of 100 is taken as the number of trees in the present study. Why 100?*

By setting the depth to infinity we indicated that we did not set a limit to the tree depth. This was a slightly misleading way to report that the "max_depth" parameter in the Scikit Learn Python module was left as default value, "None". When there is no maximum depth assigned, the algorithm will expand the regression tree nodes until the so-called leaves meet the purity criteria (mean squared error). We will rewrite some parts in Section 2.2 to clarify this point.

We tested how the maximum depth of the trees affects the model performance by altering the max_depth parameter and applying the trained model to year 2020 data (I.e., outside of training and testing phase data). The results are presented in the Figure below.

[Figure]

As Figure above shows, our RF modelling approach is not very sensitive to the max_depth parameter. Therefore, we chose not to fix the maximum depth of a tree in our modelling approach. Note that the root mean squared error and mean absolute error values are slightly larger for the year 2020 as the COVID19 pandemic caused an unusual, long-term drop in the surface PM2.5, which was not accounted for in the ECLIPSE emission inventories.

The number of trees was set to 100 based on a trial-and-error approach. There was no significant improvement if we increased the number of trees (See Table above). Due to these reasons, the hyper parameters were not adjusted based on tight cross-validation routines, but were selected near the recommended, default values.

• *Why the feature importance values are normalized, by doing so what is the chance of smoothing the non-linearity of the dependence of the input variables? Isn't there any criteria to fix the number of input variables? As the authors have pointed out the input variables are*

*mutually correlated which is obvious in the atmosphere, including all of them may lead to over-fitting. What is the authors' claim on this point?*

The feature importance values are normalized to make them more comparable to each other. They represent the contribution of each variable on the reduction of the error criteria, and the normalization scales these to a scale from zero to one. Without normalization, the importance values would be more difficult to interpret as they would always depend on the computed total error of the particular RF training setup. Furthermore, the normalizing routine is a default setting in the applied SciKit Python library.

It is true that, since there are correlated input variables, there is a small risk of overfitting. We did preselection for the input variables based on feature importance values. We excluded variables that had an average feature importance value close to zero. Furthermore, for each regression tree, the maximum number of input features to be used when looking for the optimal split was fixed to 7 (See Section 2.2), so that there is a randomized set of features for splits. However, we did not carry out a stringent optimization routine to prune the number of input variables to as low as possible. This was because we had 31 stations and thereby 31 RF models. Each of the stations has distinct characteristics, and therefore the optimal set of input variables is slightly different for each station. We estimated that using the same set of input variables for all RF models could result in a more harmonized outcome. In order to minimize the risk of overfitting, we will re-evaluate the set of input features and analyze the impact of highly correlated input features on the RF predictions.

Furthermore, we will update the Section to describe more explicitly how the input features were selected. In addition, we noticed that there were some inconsistent descriptions of the modelling setup, and we will update those to correspond to our modeling parameters.

We further noticed that the description of the algorithm implementation slightly differed from the actual modelling setup. We will therefore update the sentence describing the "max_features" property. In addition, the bootstrap bagging method was not applied, and therefore we will remove the sentence indicating that. We further tested the model performance with different bootstrapping parameters, and there was no detectable improvement whether the bootstrapping was applied or not.

• *For the global-scale modelling, ECLIPSE V6b emission scenarios are used. How appropriate is this emission inventory for simulating PM over the Indian region or what are the criteria for selecting this inventory for this study? What is the contribution of this inventory to the high under-estimation of PM loading by the model over Delhi as shown in the manuscript?*

A detailed description of previous versions of the ECLIPSE inventories is presented in Klimont et al. (2017) and Stohl et al. (2015).

The ECLIPSE V6b emissions are described in detail in the forthcoming Arctic Monitoring and Assessment Program (AMAP) report. They were especially developed to study emission reductions of short-lived climate forcers (SLCF) on the global and Arctic climate. Unfortunately, the publication of the report has been put on hold due to the current political situation for an indefinite amount of time. However, Belis et al. (2022), von Salzen et al. (2022) and Whaley et al. (2022) provide brief descriptions of the ECLIPSE V6b scenarios.

Furthermore, as we mention in the manuscript text lines 449-451, Whaley et al. (2022) state that in ECLIPSE V6b, the recent declines in Asian SO2 and BC emissions are considered. This suggests that ECLIPSE V6b is a better choice for modeling South Asian aerosols, since some emission inventories, such as CEDS emissions from CMIP6 simulations, might lack this decline (Wang et al. 2021).

We will update the manuscript text to include additional citations for both ECLIPSE inventory and the GAINS model.

• *How the exclusion of the mineral dust component solves the issue related to the PM2.5 peaking? How authors can make sure that this exclusion won't affect the other simulation results?*

Here we want to clarify the RF-correction procedure. Instead of using RF to directly predict surface PM2.5 values, we predict a correction term to the PM2.5 values modelled by ECHAM-HAMMOZ, which has been shown earlier to give better results (e.g., Lipponen et al., 2013). For the computation of the correction term, we use all the ECHAM-HAMMOZ parameters listed in Table 3, which does also include the PM2.5 due to mineral dust component and mineral dust emissions from ECHAM-HAMMOZ. However, because during the RF training phase the mineral dust component shows very large peaks which do not correspond to the measurements, we exclude mineral dust from the PM2.5 value to which the correction term is added. Therefore, the correction term includes an inferred amount of mineral dust as the random forest models get information about mineral dust episodes as an input. In this sense, the mineral dust component is not (entirely) excluded from the RF-correction procedure. We will clarify this in the manuscript.

The difference between PM2.5 with and without mineral dust can be seen in Figure S2. Most of the very high peaking values in ECHAM-HAMMOZ data are during summertime, and due to mineral dust. The daily average of the measurement stations, on the other hand, does not show summertime peaking values that would exceed the winter month maxima. Including dust component in the ECHAM-HAMMOZ PM2.5 might have produced the highest values for the summer months, as our bias correction is additive. We agree with the referee that excluding mineral dust from ECHAM-HAMMOZ PM2.5 when calculating of the error term between station PM2.5 and ECHAM-HAMMOZ might affect results to some level as the very short-term peaks might be suppressed. However, our aim was to model the long-term effects of aerosol

mitigation, and that is why we chose to prefer improved seasonal trends over short-term minima and maxima values.

We will add text in Section 2.2 to describe why avoiding dust peaks was necessary.

• *Coming to the radiative forcing calculations (section 2.8), how is the definition given to the radiative forcing is related to the conventional definitions found in the published literature? If there is any difference please highlight and justify those, else give supporting citations.*

Thanks for the suggestion, we will insert additional citations in Section 2.8. For the calculation of $RF_{ARI}$, we have described in the manuscript text the small differences between our definition and the conventional definition published in the 5th IPCC assessment report. The ERF calculations are according to the latest IPCC assessment report (see answer below).

• *How the effective radiative forcing is estimated?*

The ERF values were calculated as described in Section 2.8, and follow the definition proposed in, for instance, the latest IPCC Assessment report (Forster et al., 2021). The ERF is the difference between the top of atmosphere (TOA) net radiative fluxes of a perturbed (MITIG_2030, CLE_2030) and reference (PRES) simulation. All the simulations have fixed sea surface temperatures (SST) and sea ice cover (SIC), and the meteorology is allowed to evolve freely, i.e., no nudging was applied.

As we mentioned above, we will add more references in Section 2.8. Furthermore, we will clarify the explanations in Section 2.8 to explicitly mention the fixed SST and SIC.

*Other comments*

*L1: This opening sentence is misleading. The study demonstrates the potential of the ML technique in downscaling a global-scale mode output..*

We will update the opening sentence to be more precise.

*L6: You mean the model output is better than the measured PM2.5 values?*

Thanks for pointing this out, the sentence was slightly vague. We'll rewrite this sentence.

*L11: This is a highly impactful statement. Better to give caution to the reader by mentioning the associated large uncertainty as seen in Fig. 3(e).*

Thank you for the suggestion. We will extend this sentence to also mention the uncertainty related to the ERF values estimated.

*L38-40: As per the sentence, the role of ACI in aerosol indirect effects is undermined, hence please modify the sentence. Also, please explain the 'local meteorological dynamics' with references.*

We will modify this sentence to as suggested.

*L72-73: Cannot find in any of the given references that 'emissions from New Delhi' significantly contribute to the ATAL. Please clarify.*

This statement was slightly misleading, as Fairlie et al. (2020) referred to Indian subcontinent and Northern India as significant emission sources contributing to the ATAL. We will correct the sentence to refer to the whole Indian subcontinent.

*L94: HAM 'threats' the chemical compounds..?*

Many thanks for pointing out this typing error. We will fix this in the manuscript text.

*L158: Why do OC emissions increase by 2030 in CLE scenario while all others show a reduction?*

In CLE scenario for the year 2030, the *global* OC emissions from waste sector increase approximately by 900 kt (+44 %) when compared to 2015 levels. Furthermore, emissions due to agricultural waste burning increase by ~200 kt (+10%) compared to year 2015 emissions, and the emissions from industry and shipping are also projected to increase by a small portion. These changes are almost as large as emission reductions in the domestic (~ -960 kt) and traffic (~ -230 kt) sectors. The net change in global anthropogenic OC emissions is a small decrease (~ -0.07 %) in the year 2030 compared to the year 2015.

For the *area surrounding New Delhi*, the increasing OC emissions from the waste (~ +90 kt, +62%) and industry (+11kt, +83%) sectors counterbalance some of the emission reductions cuts from the domestic (-38kt) and traffic (-18kt) sectors. Therefore, there is an increase in the net OC emissions in the area surrounding New Delhi. For BC and SO2, the reductions in other sectors (domestic and traffic for BC and energy sector for SO2) are large enough to balance out the increasing emissions from waste sector. That is why the 2030 CLE emissions for BC and SO2 are less than in 2015.

We will add a few sentences to the manuscript text to explain briefly the trends in emissions for

different sectors.

*L267: What is '2D yearly mean value'?*

By 2D, we meant that the yearly mean values were calculated separately for each grid box. We will rewrite this in a more transparent manner.

*L356: If that is the case, what is the significance of feature importance values?*

The feature importance values describe which input features reduced the modeling error the most in the training phase. Therefore, the importance values reveal information about the RF algorithm priorities during the training. However, the feature importances do not necessarily describe the RF model output sensitivities to input features. We will append this sentence to the manuscript to describe this difference better.

*L385: This _ _ somewhat?*

We will modify this sentence to clarify the main point, I.e., that the strong negative RF$_{ARI}$ values were not expected when all three aerosol species are reduced simultaneously.

*L395-398: Confusing. The aerosol loading in MITIG_2030 is supposed to be lower than the CLE_2030, then how RF in MITIG_2030 is more negative than CLE_2030 at the Himalayan foothills. Bright background due to strong haze is expected to be more in CLE_2030.*

The RF$_{ARI}$ describes the change in the aerosol radiative effect between perturbed and reference scenarios. MITIG_2030 RF$_{ARI}$ is more negative than CLE_2030 RF$_{ARI}$ since there is less absorbing aerosol (BC) in MITIG_2030 than in CLE_2030. It is true that the pollution haze is expected to be stronger in CLE_2030 than in MITIG_2030. However, as we see from Figures S3b and d, the change in atmospheric absorption is significantly less in MITIG_2030 than in CLE_2030. This indicates that the absorption due to BC is dominating effect in this area, and therefore the RF$_{ARI}$ is more negative in MITIG_2030 than in CLE_2030. We will modify the sentence slightly to describe this mechanism better.

*L434-436: cannot understand. Do you mean that the CDNC burden was more in CLE_2030 scenario?*

Exactly. We will rewrite the sentence to make this point clear.

*L435: Expand CDNC in the manuscript.*

Thanks for pointing out this deficiency. We will add a clarification of the abbreviation and will also describe the term "burden" in Section 2.6.

*Conclusions: This section can be renamed as summary and conclusions by adding the significant findings of the study as bullet points.*

Many thanks for the suggestion, we have renamed Section 4 and will modify the text to fit the new naming.

**References:**

Allen, R. J., Turnock, S., Nabat, P., Neubauer, D., Lohmann, U., Olivié, D., Oshima, N., Michou, M., Wu, T., Zhang, J., Takemura, T., Schulz, M., Tsigaridis, K., Bauer, S. E., Emmons, L., Horowitz, L., Naik, V., van Noije, T., Bergman, T., Lamarque, J.-F., Zanis, P., Tegen, I., Westervelt, D. M., Le Sager, P., Good, P., Shim, S., O'Connor, F., Akritidis, D., Georgoulias, A. K., Deushi, M., Sentman, L. T., John, J. G., Fujimori, S., and Collins, W. J.: Climate and air quality impacts due to mitigation of non-methane near-term climate forcers, Atmos. Chem. Phys., 20, 9641–9663, https://doi.org/10.5194/acp-20-9641-2020, 2020.

Belis, C. A., Van Dingenen, R., Klimont, Z., and Dentener, F.: Scenario analysis of PM2.5 and ozone impacts on health, crops and climate with TM5-FASST: A case study in the Western Balkans, Journal of Environmental Management, Volume 319, 115738, ISSN 0301-4797, https://doi.org/10.1016/j.jenvman.2022.115738, 2022.

Fairlie, T. D., Liu, H., Vernier, J.-P., Campuzano-Jos,t, P., Jimenez, J. L., Jo, D. S., Zhang, B., Natarajan, M., Avery, M. A., and Huey, G.: Estimates of Regional Source Contributions to the Asian Tropopause Aerosol Layer Using a Chemical Transport Model, Journal of Geophysical Research: Atmospheres, 125, e2019JD031 506, https://doi.org/https://doi.org/10.1029/2019JD031506, 2020.

Forster, P., Storelvmo, T., Armour, K., Collins, W., Dufresne, J.-L., Frame, D., Lunt, D.J., Mauritsen, T., Palmer, M.D., Watanabe, M., Wild, M. and Zhang, H.: The Earth's Energy Budget, Climate Feedbacks, and Climate Sensitivity. In Climate Change 2021: The Physical Science Basis. Contribution of Working Group I to the Sixth Assessment Report of the Intergovernmental Panel on Climate Change [Masson-Delmotte, V., P. Zhai, A. Pirani, S.L. Connors, C. Péan, S. Berger, N. Caud, Y. Chen, L. Goldfarb, M.I. Gomis, M. Huang, K. Leitzell, E. Lonnoy, J.B.R. Matthews, T.K. Maycock, T. Waterfield, O. Yelekçi, R. Yu, and B. Zhou (eds.)]. Cambridge University Press, Cambridge, United Kingdom and New York, NY, USA, pp. 923–1054, doi:10.1017/9781009157896.009, 2021.

Hastie, Trevor., et al. The Elements of Statistical Learning : Data Mining, Inference, and Prediction. 2nd ed., Springer, pp. 590. 2009.

Klimont, Z., Kupiainen, K., Heyes, C., Purohit, P., Cofala, J., Rafaj, P., Borken-Kleefeld, J., and Schöpp, W.: Global anthropogenic emissions of particulate matter including black carbon, Atmos. Chem. Phys., 17, 8681–8723, https://doi.org/10.5194/acp-17-8681-2017, 2017.

Lipponen, A., Kolehmainen, V., Romakkaniemi, S., and Kokkola, H.: Correction of approximation errors with Random Forests applied to modelling of cloud droplet formation, Geoscientific Model Development, 6, 2087–2098, https://doi.org/10.5194/gmd-6-2087-2013, 2013.

Stohl, A., Aamaas, B., Amann, M., Baker, L. H., Bellouin, N., Berntsen, T. K., Boucher, O., Cherian, R., Collins, W., Daskalakis, N., Dusinska, M., Eckhardt, S., Fuglestvedt, J. S., Harju, M., Heyes, C., Hodnebrog, Ø., Hao, J., Im, U., Kanakidou, M., Klimont, Z., Kupiainen, K., Law, K. S., Lund, M. T., Maas, R., MacIntosh, C. R., Myhre, G., Myriokefalitakis, S., Olivié, D., Quaas, J., Quennehen, B., Raut, J.-C., Rumbold, S. T., Samset, B. H., Schulz, M., Seland, Ø., Shine, K. P., Skeie, R. B., Wang, S., Yttri, K. E., and Zhu, T.: Evaluating the climate and air quality impacts of short-lived pollutants, Atmos. Chem. Phys., 15, 10529–10566, https://doi.org/10.5194/acp-15-10529-2015, 2015.

von Salzen, K., Whaley, C.H., Anenberg, S.C. et al: Clean air policies are key for successfully mitigating Arctic warming. Commun Earth Environ 3, 222. https://doi.org/10.1038/s43247-022-00555-x, 2022

Whaley, C. H., Mahmood, R., von Salzen, K., Winter, B., Eckhardt, S., Arnold, S., Beagley, S., Becagli, S., Chien, R.-Y., Christensen, J., Damani, S. M., Dong, X., Eleftheriadis, K., Evangeliou, N., Faluvegi, G., Flanner, M., Fu, J. S., Gauss, M., Giardi, F., Gong, W., Hjorth, J. L., Huang, L., Im, U., Kanaya, Y., Krishnan, S., Klimont, Z., Kühn, T., Langner, J., Law, K. S., Marelle, L., Massling, A., Olivié, D., Onishi, T., Oshima, N., Peng, Y., Plummer, D. A., Popovicheva, O., Pozzoli, L., Raut, J.-C., Sand, M., Saunders, L. N., Schmale, J., Sharma, S., Skeie, R. B., Skov, H., Taketani, F., Thomas, M. A., Traversi, R., Tsigaridis, K., Tsyro, S., Turnock, S., Vitale, V., Walker, K. A., Wang, M., Watson-Parris, D., and Weiss-Gibbons, T.: Model evaluation of short-lived climate forcers for the Arctic Monitoring and Assessment Programme: a multi-species, multi-model study, Atmos. Chem. Phys., 22, 5775–5828, https://doi.org/10.5194/acp-22-5775-2022, 2022.

---

## Author Response (AR1)

**Responses to the referee comments:**
**Assessing the climate and air quality effects of future aerosol mitigation in India using a global climate model combined with statistical downscaling**

*Miinalainen, T., Kokkola, H., Lipponen, A., Hyvärinen, A.-P., Soni, V. K., Lehtinen, K. E. J., and Kühn, T.: Assessing the climate and air quality effects of future aerosol mitigation in India using a global climate model combined with statistical downscaling, Atmos. Chem. Phys. Discuss. [preprint], https://doi.org/10.5194/acp-2022-513, in review, 2022.*

We thank Anonymous Referees for the efforts made to review our paper and for the valuable comments. Our responses are written below each comment separately. The referee comments are marked with *yellow color and italic*, and the author replies are marked with gray color. The original manuscript text is marked with pink color, and updated text with dark magenta. The line numbers refer to the original, submitted version of the manuscript.

**Replies to the comments made by the Anonymous Referee #1:**

*This manuscript presents global model simulation results over India from ECHAM for air quality and radiative forcing under present and future emission scenarios (from GAINS) through 2030. For the region covering Delhi, the results were downscaled using random-forest corrections using multiple emission, met, and orography variables.*

*While the radiative forcing calculations from ECHAM is a known path, the use of the same to bias correct and estimate air quality for a city is new. The later methods have been used, but not for model resolutions at 1.9 degrees. Machine Learning (ML) approach is a new and emerging field and the benefits of using a global model for both air quality and climate applications cannot be overlooked. While the methodology is well explained for correcting the model results with biases from ML, the statistics also improved after the corrections, the gaps between the measured and model-corrected numbers is still significant.*

We thank Anonymous Referee #1 for the useful comments.

As the referee mentioned, the RF-corrected PM2.5 does not fully reproduce the station-averaged PM2.5 for the testing phase data, especially over short time scales. This is analyzed and discussed in Section 3.1.

However, if the aim is to capture the long-term trends correctly, the RF-corrected PM2.5 performs as adequately as the PM2.5 obtained from the dispersion model System for Integrated modeLling of Atmospheric coMposition (SILAM). In the Table below, we show the comparison statistics between modelled and mean of the New Delhi ground stations for the testing phase period 22.8.2018-31.12.2019.

| Compared to the mean of the measurement station PM25 for 22.8.2018-31.12.2019, daily average values | | | |
|---|---|---|---|
| | ECHAM PM2.5 | SILAM PM2.5 | RF-corrected PM2.5 |
| RMSE (µg/m³) | 113 | 65 | 56 |
| MRE (%) | -36 | 27 | 33 |
| MAE (µg/m³) | 78 | 46 | 39 |
| R² | -0.63 | 0.46 | 0.59 |
| Pearson correlation | 0.19 | 0.79 | 0.79 |

The error statistics show that the RF-corrected PM2.5 correlates with the measured data at the same level as SILAM PM2.5 does. Furthermore, the mean absolute error (MAE) and root mean squared error (RMSE) are smaller for RF-corrected PM2.5 than for SILAM data. The mean relative error (MRE) is smaller for SILAM than for RF-corrected data, but the difference is less than 10 percent units. This comparison suggests that the performance of the downscaling approach is comparable to air quality models if the aim is to utilize data to applications where long-term trends in air quality are central.

*The scenario analysis for air quality primarily hinges on the reproductive capacity of the model and the only question that is not clearly answered is why extract air quality data from such a coarse model (when the problem is known that coarser models have hard time replicating high-density urban areas with very distinct emission characteristics)? Especially, since FMI and IMD (author organizations) are known to conduct chemical transport modeling for air quality at much better resolutions globally and in India.*

The referee is correct that the downscaled PM2.5 does not fully correspond to outputs from high resolution models when comparing day-to-day variation. However, this was not the aim of this project, and we presume that the concept of our manuscript was misinterpreted by the referee.

The main idea of our study is to expand the possibilities to utilize global model data in additional applications such as local air quality analysis. As we describe in the manuscript text (line 522 and forward), one advantage of using global-scale models for analyzing the effects of aerosol mitigation is that one can simulate fairly long time periods (decades to even a century), and that the simulations typically cover the whole globe. This is especially relevant for future projections of climate and air quality. With the help of downscaling, one can "zoom in" to a

very specific location and analyze how the global or local scale emission mitigation affects. Up to our understanding, air quality models are computationally more expensive due to the high grid resolution, and therefore the simulated time periods are shorter than with GCMs, and the simulation times can be much longer.

Furthermore, another advantage of using downscaling enhanced PM2.5 from a GCM is that one can simultaneously analyze the effects of aerosol mitigation on various other climatic processes. In our case, we focused on the radiative forcing, but one could, for instance, analyze global precipitation patterns or low-level cloud formation at very distinct regions in the globe.

One plausible application of downscaling PM2.5 could be simply to use it as a "quick tool" to evaluate how a mitigation scheme affects future climate or air quality in different parts of globe. The intention of this study is not to present a method which can be used to replace chemical transport models. Instead, the main idea is to provide a relatively light-weight tool that can be used to assess simultaneously the climate and air quality effects of future aerosol emission reduction measures.

Moreover, air quality models require simulated data from global climate models for modelling the future climate. Regional modelling with fine grid resolution requires input for boundary conditions for large scale atmospheric dynamics, and therefore a separate global model simulation is needed before the actual air quality model simulation can be performed. Our intention was to explore if we could use the global model data directly also to analyzing surface air quality, especially if there is no need to study the underlying physical mechanisms in a detailed way.

We have updated the Introduction text by adding the following sentences to line 71:
"The aim of this work is to explore more versatile methods to utilize global climate model data. The combination of ECHAM-HAMMOZ and statistical downscaling could be used as a relatively light-weight tool for applications where one aims to assess the effects of different aerosol mitigation strategies simultaneously on larger scale climate, and on local air quality."

On line 54 we added:
"Air quality models, on the other hand, are not well suited to assess climate impacts of emission changes. Furthermore, when modelling future emission scenarios, air quality models require input from global climate models to set boundary and initial conditions."

*Why use a city like Delhi with so many stations with 0% data available in the ML testing phase?Why not use a city in Europe or the US with good availability rates and good representation of the sources, to show that the model is capable of replication after the bias corrections?*
We assume this is a misunderstanding as the majority of the stations used in this study have a coverage of 80 to 90% for the testing phase (see Table S1).

To emphasize this point, we have modified the text at lines 141-142 from
"the final RF bias correction leans more on the year 2018 (and 2019) data since there is a higher number of measurement points for those years." to
"the final RF bias correction leans more on the year 2018 data since there is a higher number of measurement points for that year."

and added a sentence after line 147:
"All in all, we had good temporal coverage for the measurement station data for both training and testing phases."

*The one drawback of the manuscript is the selection of the case study city (Delhi) -- which has strong seasonal trend, strong diurnal trend, and distinct sources (for SO2, BC, and OC) over the months. A city(s) or region(s) with consistent emission loads would cut down some uncertainty in the model and corrections methods and then apply to regions like India and China.*

We chose New Delhi as a case study city because of its diurnal and seasonal trends, as it is ideal for this kind of study. The PM2.5 concentrations are at a high level for most of the year in New Delhi. Furthermore, Delhi National Capital Territory (NCT) is densely populated and relatively large in surface area. In addition, the ground measurement station network in New Delhi is very extensive and there was a sufficient amount of data available for our study.

Furthermore, we respectfully disagree with the referee that the unique characteristics backgrounding New Delhi pollution profile would make Delhi unsuitable for this kind of analysis. On the contrary, as we mention right in the first sentence of the abstract, we aimed to study the potential of the downscaling procedure. As the referee described, the New Delhi PM2.5 is not constant all year around but has a lot of seasonal variation and strong dependence on anthropogenic sources. This makes New Delhi a good target region to explore how well downscaling can capture local tendencies, such as short- and long-term trends in PM2.5.

Anthropogenic aerosol emissions are relatively large in India (~15% of the global BC emissions), and they are not projected to decrease at the same rate as global emissions (see manuscript Table 1). That is why we considered that India and New Delhi are interesting areas to study, as the aerosol mitigation is expected to bring clear benefits to the local air quality, but the net radiative forcing due to simultaneous mitigation of three species (BC, OC and SO2) was unclear. We have updated the Introduction text to give readers a clear impression of why New Delhi was chosen as a study of interest by adding at line 80:
"In addition, New Delhi PM2.5 shows strong seasonal variation and high dependence on anthropogenic emissions (Bawase et al., 2021). This makes New Delhi an ideal target region to study how well downscaling can capture short- and long-term trends in PM2.5."

We have also added after line 127:
"The ground measurement station network in New Delhi is extensive and there was a good

amount of measurement data available, which made New Delhi ideal location for this kind of study."

Unfortunately, it seems that we have not reported clearly enough in our manuscript that our bias correction model is meant to be built separately for each city. Though technically possible, it would require a very large global training dataset representing a wide variety of conditions to construct an ML-based model that would be able to generalize from one region to another. In practice, this type of global dataset does not exist. The characteristics of PM2.5 concentrations trends and how they depend on prevailing atmospheric conditions and the assumed local emissions are unique for each city/region. We used the same city for training and testing the RF Model to ensure representative training data for the model.

We have emphasized this in Section 2.7 by adding after line 247:
"One should keep in mind that in this bias correction approach, the models are assumed to be city specific. In other words, RF models trained with New Delhi data are not meant for correcting biases in ECHAM-HAMMOZ PM2.5 for other cities."

*Line 237-242 and 290: It is not clear if the emissions and other variables extracted and used are still at the ECHAM resolution or further downscaled to support a region of 30km x 30km over Delhi? (L290) is an important observation - When making the bias corrections, besides the model grid variables, are there any variables that are seggregating the Delhi area signatures for a better fit?*
We thank the referee for making us aware that this aspect was not clear enough in the manuscript text. The input feature values used in the RF bias correction are all from one specific ECHAM-HAMMOZ simulation, and the data were not further downscaled to represent higher resolution grid data. Downscaling from T63 to L290 would require a large amount of external fine resolution data and would not serve the purpose of lowering the computational costs.

Instead, we post processed the input feature data by extracting values from the original ECHAM-HAMMOZ data of T63 horizontal resolution. We produced input feature data either by extracting data from one single grid box ('point'), or by summing ('fldsum') or averaging ('fldmean') over an area surrounding New Delhi (72 – 82 'W, 24 - 34 'N). This means that some of the input features represent more the regional changes in the simulated atmosphere, whereas the point features show more the local changes.

In our modeling set up, it is assumed that the measurement data from ground stations incorporate information about the very detailed characteristics affecting New Delhi pollution levels, such as urban infrastructure or very specific emission sources. Including detailed external data as an input feature (e.g., info about public traffic routes) would have reduced the feasibility of our modeling approach since external data is easily subject to change and would need to be kept updated constantly or be based on additional scenarios or assumptions. Then again, the absence of detailed, local information is most likely the reason why our modelling

approach is not capturing short-term variations very well. Developing such a downscaling bias correction method that also includes external data could be an opportunity for future studies.

We have modified the sentence at line 235 from:
"The input features used in the random forest fitting are presented in Table 3." to
"The input features used in the random forest fitting are purely ECHAM-HAMMOZ data, and we did not use any external data as input predictors. The input features are presented in Table 3."

We also added a description of how the input features were selected. We added after line 237:
"We selected the input features mainly based on the average feature importance values. At first, we formed a larger set of input features and ran the RF modelling with this larger set of input features. After that, we pruned out those variables that had an average feature importance value close to zero, i.e., they were assumed to have a very minor influence on the RF correction. Finally, we identified pairs of strongly correlated input features (with correlation coefficients larger than 0.9) and removed one of the two features."

In addition, we added after line 521:
"Furthermore, the method we proposed could be further advanced by including very detailed information about local infrastructure and traffic statistics, for instance. This could potentially improve the correction of short-term trends in New Delhi PM2.5."

*The results and conclusions of the study in terms of AQ and climate benefits of reducing emissions are as expected. However, since Delhi is the most polluted area/city in the world with not only a complex mix of emission sources, but also a complex mix of political and instititutional setup to manage these emissions. While the manuscript presented % changes (benefits for air quality and RF), the discussion doesn't include any explanation on how these % emission reductions will be acheived in the Delhi area. It is understood that the emissions work comes from a different model (GAINS). Since the manuscript very specifically mentions and analyzes data for one city only, it would be appropriate to also discuss this space.*

We would like to highlight here that it is not explicitly evident that future aerosol mitigation would bring a net negative forcing over India. As many studies have shown (for instance, Allen et al.,2020), simultaneous reductions in both scattering and absorbing aerosols are expected to reduce the net cooling effect of aerosols on a global scale, thereby inducing a warming contribution.

The aerosol mitigation scenarios used in this study were part of the ECLIPSE V6b emission scenarios, which were designed with the GAINS model. The underlying assumptions in ECLIPSE are idealized in the sense that emission reductions are based on assumed perfect achievement of currently valid legislation (CLE) or full implementation of all currently available technologies to reduce emissions (MFR). In this sense these scenarios could be seen as maximum possible emission reductions from the 2020 viewpoint. Analyzing the underlying political actions needed

to achieve the proposed emission reductions is beyond the scope of this study. A more detailed description of the ECLIPSE scenarios can be found from e.g. Stohl et al. (2015), Belis et al. (2022) and Klimont et al. (2017).

However, we have added after line 162:
"Both globally and in the New Delhi surroundings, the sectors contributing most to reductions in BC emissions were the domestic and traffic sectors, while for OC, the domestic and waste sectors had the most significant reductions. For $SO_2$, the largest reductions were projected in the MFR scenario for the energy and industry sectors."

In addition, we have extended the discussion at Section 3.2 by adding after line 329:
"As described in Section 2.2, the largest reductions for anthropogenic aerosol emissions under future mitigation scenario were assumed to come from domestic (BC & OC), waste (OC), traffic (BC), industry ($SO_2$) and energy ($SO_2$) sectors. The emissions from different sectors contribute differently to the atmospheric aerosol concentrations, because the assumed size distributions, mixing states and injection height at emission time are different. Each sector therefore also contributes differently to the modelled PM2.5 values."

*While there is merit to a new methodology to be able to model AQ data along with the climate data, the manuscript lacks punch and I am afraid that these correction results will be hard to replicate in another setting.*

While we appreciate the referee's feedback, we respectfully disagree that the proposed method would not be applicable for analyzing the effects in another city or region if trained and applied with suitable data. We do, however, recognize that the performance of the bias correction may be different for each target city.

As mentioned above, there might be a misunderstanding in that we would propose to apply the model trained with New Delhi data to correct air quality levels for a different city. In case this referee's comment is based on that kind of assumption, we have elucidated this in the manuscript text as mentioned above in one of the answers.

In addition to the already mentioned improvements, we have gone through the whole manuscript text and highlighted the core concepts to sharpen the key messages of our study.
* * *
**Replies to the comments made by the #Anonymous referee 2:**

*Review of "Assessing the climate and air quality effects of future aerosol mitigation in India using a global climate model combined with statistical downscaling"*

*The manuscript explores the possibilities of using a global climate model to investigate the effects of aerosol mitigation in India. A machine learning (ML) approach using Random Forest regression is used to downscale PM2.5 concentrations over a polluted city, New Delhi with the help of measured PM2.5 concentrations. Different PM loading future scenarios are projected and compared with the uncorrected and ML-corrected model outputs. The effects of aerosol mitigation are investigated in terms of radiative effects and effective radiative forcing under the PM future scenarios. The authors claim the improvement of global-scale model output in simulating the PM2.5 concentration over a small domain and their effectiveness in estimating the radiative impacts. The study demonstrates the potential of the emerging technique of ML in improving the large-scale model output in the process of statistical downscaling. The study is relevant and unique as mentioned above, and has a significant contribution to the relevant scientific domain. However, some concerns remain significant and need to be considered before publishing.*

*General Comments*

*The manuscript focuses on two aspects. (1) Demonstration of an ML technique in improving a global-scale model to simulate the PM2.5 concentrations over a small region via statistical downscaling under different emission scenarios. (2) Estimating the radiative effects of future aerosol scenarios using the RF-corrected model simulations. The manuscript structure is difficult to follow until reaching the present 'Conclusions' section which is not a conclusion, but a nice overview/summary of the entire work.*

We thank the referee for the insightful suggestions to improve the manuscript structure and content. We have re-organized some of the subsections in Section 2 and renamed Section 4 as suggested.

*If the authors want to highlight their simulation results regarding the impact of future aerosol mitigation, more discussion is needed with proper references to the existing findings, else it remains as a technical paper demonstrating the potential of ML in statistical downscaling. Currently, the physical mechanisms for some of the simulation outcomes are not given/found, but some tentative reasons are proposed. Many studies are documented the current aerosol-impact scenario using multiple scientific techniques (in situ, remote sensing, etc.) and future projections also for the Indian region.*

We have added to Section 3.2. after line 351 the following:
"Moreover, we compared the average regional convective precipitation values for the New Delhi surroundings (see Section 2.6). The average convective precipitation for June and July was $(4.4 \pm 4.1) \times 10^{-5}$ mm s$^{-1}$ and $(4.2 \pm 4.1) \times 10^{-5}$ mm s$^{-1}$ for PRES and CLE_2030, respectively. For MITIG_2030, the average summertime convective precipitation was $(3.1 \pm 3.8) \times 10^{-5}$ mm s$^{-1}$. Based on the Mann-Whitney U-test, there was a statistically significant difference between MITIG_2030 and PRES summertime precipitation in the New Delhi surroundings. Decreased

summertime convective precipitation indicates less washout, allowing for higher surface PM2.5 concentrations in the MITIG_2030 simulation."

We have added to line 426:
"The areas for which we observed the largest changes in atmospheric SW absorption are almost the same regions as obtained by Prakash et al. (2020), even though their analysis handled only the BC emissions due to road transport."

Furthermore, we have added to Section 3.3., after line 410:
"Das et al. (2020) reported an increase in dust burden over central India, due to decreases in summertime precipitation. Our results also showed a decrease in convective precipitation when comparing MITIG_2030 values to PRES (see Section 3.2), which may have contributed to the increased in dust loadings in MITIG_2030. However, the modelled natural variation in precipitation patterns is large, and therefore there is significant uncertainty related to this mechanism."

*To ascertain the second aspect of the current manuscript, the first part needs to be flawless and should be explained confidently. Many parts of the manuscript are confusing which calls for further explanations for the smooth reading.*
We thank the referee for bringing up that some parts of the manuscript may be confusing to the reader. We have gone through the entire manuscript and done our best to improve the general readability of the text.

*One of the highlights of the findings is that the improvement of air quality is mostly due to the reduction of OC loading. However, the negative radiative forcing is attributed to the reduction of BC emission. This is an example of confusion arising while going through the manuscript.*
This was an excellent comment and helped us to understand which parts of our manuscript are confusing. The reason why OC influences air quality more than BC, though BC has stronger effects on the radiative balance, is related to the optical properties of these aerosol species. The atmospheric mass load of BC is relatively small compared to the mass load of OC. The radiative forcing per mass load of BC, on the other hand, is much bigger than for OC, as the optical properties differ. That is why in this study the emission reductions of OC influence PM2.5 concentrations more, while BC-reductions have a larger impact on radiative forcing. Due to the strong absorptive nature of BC, the negative effect comes from reduction in absorption of shortwave radiation which outweighs the radiative effect of OC which would lead to a positive radiative effect. Permadi et al., (2018), who studied the co-benefits of BC emission reductions on air quality and radiative forcing over South East Asia also found a similar decrease in BC radiative forcing when studying emission reductions in Indonesia and Thailand. They also concluded that this negative radiative effect is due to decrease in absorption of shortwave radiation by BC. This is now discussed in the revised manuscript

We have updated Section 3.2. by adding to line 329:
"Table 1 shows that the total OC emission mass is more than double the mass of emitted BC, both on a global and regional scale. Furthermore, the contribution of sulfate to the long-term PM2.5 in New Delhi is known to be lower than the contribution of OC (Lalchandani et al., 2022), though sulfate might have a more significant role during winter haze periods."

Furthermore, we clarified Section 3.3 discussion by adding to line 398:
"While OC contributes strongest to the surface PM2.5 loadings (see Section 3.2), BC makes the strongest per unit mass contribution to the radiative forcing and hence can outweigh the cooling effects of OC and SO4, especially over bright backgrounds."

We also added after line 431:
"Permadi et al., (2018) conducted a similar study for BC emissions reductions in Indonesia and Thailand. Their outcome was that along improved air quality, emissions reductions led to a decrease in absorption of SW radiation by BC, and therefore the resulting radiative forcing was negative."

*The map of India shown in Fig. 3 is not matching with the maps published by the institutions that provided the insitu data nor with one of the authors' affiliated institution. Please correct the map as per the source or remove the political boundaries as per the journal's recommendations.*
We thank the referee for pointing out this issue. We have removed the political boundaries from Figures 3 and S3.

*Language also needs improvement. The main concerns are listed below.*

*Methodology:*

*• Why does section 2.2 stand apart from sections 2.6 and 2.7?*

We have moved Sections 2.2 and 2.3 below Section 2.5 and updated the numbering of the subsections.

*• The hyper parameters were adjusted using different combinations based on the best error statistics. Can you please show the performance of the validation test data?*

This statement in the manuscript was somewhat inaccurate as the hyper parameter selection seemed not to affect the model performance substantially. The error statistics for testing different RF parameter combinations are presented in the Figure below.

| max_depth: None, Max_features = 7 | nr of trees | 20 | 50 | 100 | 200 | 300 |
|---|---|---|---|---|---|---|
| | RMSE (µg/m³) | 56.56 | 56.52 | 56.43 | 56.52 | 56.54 |
| | MRE (%) | 32.29 | 31.30 | 31.98 | 32.21 | 32.26 |
| | MAE (µg/m³) | 38.69 | 38.46 | 38.57 | 38.57 | 38.65 |
| | $R^2$ | 0.59 | 0.59 | 0.59 | 0.59 | 0.59 |
| | Pearson correlation | 0.79 | 0.79 | 0.80 | 0.79 | 0.79 |

| max_depth: None, Max_features = None | nr of trees | 20 | 50 | 100 | 200 | 300 |
|---|---|---|---|---|---|---|
| | RMSE (µg/m³) | 56.79 | 56.81 | 56.90 | 56.84 | 56.76 |
| | MRE (%) | 33.15 | 33.52 | 33.46 | 33.08 | 33.11 |
| | MAE (µg/m³) | 39.02 | 39.09 | 39.16 | 39.09 | 39.01 |
| | $R^2$ | 0.59 | 0.59 | 0.58 | 0.59 | 0.59 |
| | Pearson correlation | 0.79 | 0.79 | 0.79 | 0.79 | 0.79 |

| max_depth: None, Max_features = 15 | nr of trees | 20 | 50 | 100 | 200 | 300 |
|---|---|---|---|---|---|---|
| | RMSE (µg/m³) | 56.73 | 56.43 | 56.45 | 56.45 | 56.53 |
| | MRE (%) | 32.15 | 32.54 | 32.63 | 33.27 | 32.62 |
| | MAE (µg/m³) | 38.57 | 38.61 | 38.69 | 38.81 | 38.75 |
| | $R^2$ | 0.59 | 0.59 | 0.59 | 0.59 | 0.59 |
| | Pearson correlation | 0.79 | 0.79 | 0.79 | 0.79 | 0.79 |

| max_depth: 10, Max_features = 7 | nr of trees | 20 | 50 | 100 | 200 | 300 |
|---|---|---|---|---|---|---|
| | RMSE (µg/m³) | 56.78 | 56.78 | 56.59 | 56.54 | 56.51 |
| | MRE (%) | 31.72 | 32.39 | 32.58 | 31.78 | 31.91 |
| | MAE (µg/m³) | 38.70 | 38.77 | 38.77 | 38.56 | 38.57 |
| | $R^2$ | 0.59 | 0.59 | 0.59 | 0.59 | 0.59 |
| | Pearson correlation | 0.79 | 0.79 | 0.79 | 0.79 | 0.79 |

| max_depth: 5, Max_features = 7 | nr of trees | 20 | 50 | 100 | 200 | 300 |
|---|---|---|---|---|---|---|
| | RMSE (µg/m³) | 56.93 | 57.00 | 57.12 | 57.09 | 57.12 |
| | MRE (%) | 31.65 | 31.18 | 31.80 | 31.87 | 31.76 |
| | MAE (µg/m³) | 38.77 | 38.76 | 38.99 | 38.96 | 38.96 |
| | $R^2$ | 0.58 | 0.58 | 0.58 | 0.58 | 0.58 |
| | Pearson correlation | 0.79 | 0.79 | 0.79 | 0.79 | 0.79 |

| max_depth: 15, Max_features = None | nr of trees | 20 | 50 | 100 | 200 | 300 |
|---|---|---|---|---|---|---|
| | RMSE (µg/m³) | 56.84 | 56.83 | 56.85 | 56.79 | 56.68 |
| | MRE (%) | 34.39 | 33.11 | 33.62 | 33.26 | 33.19 |
| | MAE (µg/m³) | 39.33 | 39.09 | 39.20 | 39.06 | 39.02 |
| | $R^2$ | 0.59 | 0.59 | 0.59 | 0.59 | 0.59 |
| | Pearson correlation | 0.79 | 0.79 | 0.79 | 0.79 | 0.79 |

As the statistics show, our RF model setup is not overly sensitive to the selection of hyper parameters. Hastie et al. (2009, p. 590) reported a similar tendency. Based on their experiences, random forests require very little tuning.

Furthermore, our modeling approach had 31 RF models (one per measurement station), and we used the same RF hyperparameters in each of them, as we did not tune each station-specific RF model separately. We therefore decided to use the default, recommended values in the RF models.

We have modified the statement at lines 111-112 from
"The hyper parameters were adjusted by testing different combinations and chosen based on the best error statistics:" to
"Based on testing different combinations, our RF model setup was not very sensitive to the choice of hyper parameters. Furthermore, we aimed to have the same hyper parameters in all of the individual RF models (see Section 2.6). Therefore, we used the recommended default values for the hyper parameters."

*• What do you mean by setting the depth of each tree to infinity? How can you make sure of avoiding over-fit while keeping the depth of the tree as infinity? What are the criteria for fixing the number of trees? It is said that a default value of 100 is taken as the number of trees in the present study. Why 100?*

By setting the depth to infinity we indicated that we did not set a limit to the tree depth. This was a slightly misleading way to report that the "max_depth" parameter in the Scikit Learn Python module was left as default value, "None". When there is no maximum depth assigned, the algorithm will expand the regression tree nodes until the so-called leaves meet the purity criteria (mean squared error).

We rewrote the sentence at lines 112-113, originally:
"The maximum depth for each tree was set to infinity and the number of trees was the default value, 100.", and after update:
"There was no maximum depth for each tree set and the depth of each tree was determined automatically based on the splitting criterion. The number of trees was the default value, 100."

We tested how the maximum depth of the trees affects the model performance by altering the max_depth parameter and applying the trained model to year 2020 data (I.e., outside of training and testing phase data). The results are presented in the Figure below.

[Figure]

As Figure above shows, our RF modelling approach is not very sensitive to the max_depth parameter. Therefore, we chose not to fix the maximum depth of a tree in our modelling approach. Note that the root mean squared error and mean absolute error values are slightly larger for the year 2020 as the COVID19 pandemic caused an unusual, long-term drop in the

surface PM2.5, which was not accounted for in the ECLIPSE emission inventories.

The number of trees was set to 100 based on a trial-and-error approach. There was no significant improvement if we increased the number of trees (See Table above). Due to these reasons, the hyper parameters were not adjusted based on tight cross-validation routines, but were selected near the recommended, default values.

*• Why the feature importance values are normalized, by doing so what is the chance of smoothing the non-linearity of the dependence of the input variables? Isn't there any criteria to fix the number of input variables? As the authors have pointed out the input variables are mutually correlated which is obvious in the atmosphere, including all of them may lead to over-fitting. What is the authors' claim on this point?*

The feature importance values are normalized to make them more comparable to each other. They represent the contribution of each variable to the reduction of the error criteria, and the normalization scales these to a scale from zero to one. Without normalization, the importance values would be more difficult to interpret as they would always depend on the computed total error of the particular RF training setup. Furthermore, the normalizing routine is a default setting in the applied SciKit Python library.

It is true that, since there are correlated input variables, there is a small risk of overfitting. We did preselection for the input variables based on feature importance values. We excluded variables that had an average feature importance value close to zero. Furthermore, for each regression tree, the maximum number of input features to be used when looking for the optimal split was fixed to 7 (See Section 2.2), so that there is a randomized set of features for splits. However, we did not carry out a stringent optimization routine to prune the number of input variables to as low as possible. This was because we had 31 stations and thereby 31 RF models. Each of the stations has distinct characteristics, and therefore the optimal set of input variables is slightly different for each station. We estimated that using the same set of input variables for all RF models could result in a more harmonized outcome.

In order to minimize the risk of overfitting, we have re-evaluated the set of input features and analyzed the impact of highly correlated input features on the RF predictions. Based on these, we have removed variables that had a correlation over 0.9 with another input feature. The excluded input variables were boundary layer height (pbl_h), OC burden (burden_OC) and PM2.5 due to OC (PM25_OC). We have repeated the RF corrections with the pruned set of input variables and updated manuscript Table 3, and also Sections 3.1 and 3.2 with the new results. Note that the differences to the original values are fairly small.

As mentioned in the responses above, we have added the following text at line 237:
"We selected the input features mainly based on the average feature importance values. At first, we formed a larger set of input features and ran the RF modelling with this larger set of

input features. After that, we pruned out those variables that had an average feature importance value close to zero, i.e., they were assumed to have a very minor influence on the RF correction. Finally, we identified pairs of strongly correlated input features (with correlation coefficients larger than 0.9) and removed the one of the two features that had a smaller average feature importance value."

Furthermore, we modified the sentence at lines 302-303 from
"As some of the input feature variables are correlated with each other, this might affect the calculated values, especially considering the random component of RF." to
"Some of the input feature variables are correlated with each other. Even though we removed input features which were correlated most strongly to other, included input features (see Section 2.6), the remaining correlated input features might affect the calculated values, especially considering the random component of RF."

In addition, we noticed that there were some inconsistent descriptions of the modelling setup, and we updated those to correspond to our modeling parameters. We noticed that the description of the algorithm implementation slightly differed from the actual modelling setup.

We updated the statement at lines 114-115 from
"The maximum number of input features to be used in one decision tree (max_features) was set to 7" to
"The maximum number of input features to be used in one decision tree node (max_features) was set to 7,"

In addition, the bootstrap bagging method was not applied, and therefore we removed the sentence indicating that. We further tested the model performance with different bootstrapping parameters, and there was no detectable improvement whether the bootstrapping was applied or not.

We have removed from lines 116-118 the statement
"In addition, we used the default bootstrap aggregation method for dividing the training data into separate sample sets in order to provide an individual sample of training data for each tree, which has been shown to increase the smoothness of the fit (Altman and Krzywinski, 2017)."

• For the global-scale modelling, ECLIPSE V6b emission scenarios are used. How appropriate is this emission inventory for simulating PM over the Indian region or what are the criteria for selecting this inventory for this study? What is the contribution of this inventory to the high under-estimation of PM loading by the model over Delhi as shown in the manuscript?

A detailed description of previous versions of the ECLIPSE inventories is presented in Klimont et al. (2017) and Stohl et al. (2015).

The ECLIPSE V6b emissions are described in detail in the forthcoming Arctic Monitoring and Assessment Program (AMAP) report. They were especially developed to study emission reductions of short-lived climate forcers (SLCF) on the global and Arctic climate. Unfortunately, the publication of the report has been put on hold due to the current political situation for an indefinite amount of time. However, Belis et al. (2022), von Salzen et al. (2022) and Whaley et al. (2022) provide brief descriptions of the ECLIPSE V6b scenarios.

Furthermore, as we mention in the manuscript text lines 449-451, Whaley et al. (2022) state that in ECLIPSE V6b, the recent declines in Asian SO2 and BC emissions are considered. This suggests that ECLIPSE V6b is a better choice for modeling South Asian aerosols, since some emission inventories, such as CEDS emissions from CMIP6 simulations, might lack this decline (Wang et al. 2021).

We have updated the manuscript text in lines 149-162 to include additional citations for ECLIPSE scenarios.

• *How the exclusion of the mineral dust component solves the issue related to the PM2.5 peaking? How authors can make sure that this exclusion won't affect the other simulation results?*

Here we want to clarify the RF-correction procedure. Instead of using RF to directly predict surface PM2.5 values, we predict a correction term to the PM2.5 values modelled by ECHAM-HAMMOZ, which has been shown earlier to give better results (e.g., Lipponen et al., 2013). For the computation of the correction term, we use all the ECHAM-HAMMOZ parameters listed in Table 3, which does also include the PM2.5 due to mineral dust component and mineral dust emissions from ECHAM-HAMMOZ. However, because during the RF training phase the mineral dust component shows very large peaks which do not correspond to the measurements, we exclude mineral dust from the PM2.5 value to which the correction term is added. Therefore, the correction term includes an inferred amount of mineral dust as the random forest models get information about mineral dust episodes as an input. In this sense, the mineral dust component is not (entirely) excluded from the RF-correction procedure.

The difference between PM2.5 with and without mineral dust can be seen in Figure S2. Most of the very high peaking values in ECHAM-HAMMOZ data are during summertime, and due to mineral dust. The daily average of the measurement stations, on the other hand, does not show summertime peaking values that would exceed the winter month maxima. Including dust component in the ECHAM-HAMMOZ PM2.5 might have produced the highest values during the summer months, as our bias correction is additive. We agree with the referee that excluding mineral dust from ECHAM-HAMMOZ PM2.5 when calculating of the error term between station PM2.5 and ECHAM-HAMMOZ might affect results to some level as the very short-term peaks might be suppressed. However, our aim was to model the long-term effects of aerosol

mitigation, and that is why we chose to prefer improved seasonal trends over short-term minimum and maximum values.

We have clarified these aspects in the manuscript text by adding after line 223: "Furthermore, due to dust episodes, the maximum PM2.5 for ECHAM-HAMMOZ was during the summer months, whereas measurement stations showed maxima for the winter season. However, mineral dust was included in the input training set (see Table 3), i.e., dust can affect the RF bias correction. PM2.5 values from ECHAM-HAMMOZ which exclude the mineral dust component are hereafter referred to as non-dust PM2.5."

*• Coming to the radiative forcing calculations (section 2.8), how is the definition given to the radiative forcing related to the conventional definitions found in the published literature? If there is any difference please highlight and justify those, else give supporting citations.*

Thanks for the suggestion, we have inserted some additional references in Section 2.8.

For the calculation of $RF_{ARI}$, we have described in the manuscript text the small differences between our definition and the conventional definition published in the 5$^{th}$ IPCC assessment report. The ERF calculations are according to the sixth IPCC assessment report (see answer below).

*• How the effective radiative forcing is estimated?*

The ERF values were calculated as described in Section 2.8, and follow the definition proposed in, for instance, the sixth IPCC Assessment report (Forster et al., 2021). The ERF is the difference between the top of atmosphere (TOA) net radiative flux of a perturbed (MITIG_2030, CLE_2030) and reference (PRES) simulation. All the simulations have fixed sea surface temperatures (SST) and sea ice cover (SIC), and the meteorology is allowed to evolve freely, i.e., no nudging was applied.

We have added a reference to the latest IPCC AR, and updated Section 2.8 to mention fixed SST and SIC, by changing at lines 264-266 from
"Here we computed the $ERF_{ARI+ACI}$ as the difference in the net radiative flux at the top of the atmosphere (TOA) between a perturbed scenario and the reference simulation." to
"Here we computed the $ERF_{ARI+ACI}$ as the difference in the net radiative flux at the top of the atmosphere (TOA) between a perturbed scenario and the reference simulation (Forster et al., 2021), using simulations with fixed SST and SIC."

*Other comments*

*L1: This opening sentence is misleading. The study demonstrates the potential of the ML technique in downscaling a global-scale mode output..*

We have updated the opening sentence from
"We studied the potential of using a global-scale climate model for analyzing simultaneously both city-level air quality and regional and global scale radiative forcing values for anthropogenic aerosols." to
"We studied the potential of using machine learning to downscale global-scale climate model output towards ground station data. The aim was to analyze simultaneously both city-level air quality and regional and global scale radiative forcing values for anthropogenic aerosols."

*L6: You mean the model output is better than the measured PM2.5 values?*

Thanks for pointing this out, the sentence was slightly vague.

We changed the sentence at line 6 from
"The downscaling procedure clearly improved the seasonal variation when compared to measured PM2.5 values." to
"The downscaling procedure clearly improved the seasonal variation of the model data. The seasonal trends were much better captured in the corrected PM2.5 than in original ECHAM-HAMMOZ PM2.5 when compared to the reference PM2.5 from the ground stations"

*L11: This is a highly impactful statement. Better to give caution to the reader by mentioning the associated large uncertainty as seen in Fig. 3(e).*

Thank you for the suggestion.

We have extended the sentence at lines 12-13 from
"This indicates that aerosol mitigation could bring a double benefit in India: better air quality and decreased warming of the climate." to
"For the two future emission scenarios modelled, the radiative forcing due to aerosol-radiation interactions over India was (−0.09 ± 0.26) W m−2 and (−0.53 ± 0.31) W m−2, respectively, while the effective radiative forcing values were (−2.1 ± 4.6) W m−2 and (0.06 ± 3.39) W m−2, respectively. Although accompanied with relatively large uncertainties, the obtained results indicate that aerosol mitigation could bring a double benefit in India: better air quality and decreased warming of the local climate"

*L38-40: As per the sentence, the role of ACI in aerosol indirect effects is undermined, hence please modify the sentence. Also, please explain the 'local meteorological dynamics' with references.*

As suggested, we have modified the text at lines 38-40 from

"In addition to these so-called aerosol-radiation interactions (ARI), aerosols can affect Earth's radiative balance indirectly, for instance, by altering the properties of clouds (aerosol-cloud interaction; ACI) or by changing local meteorological dynamics" to

"These are termed aerosol-radiation interactions (ARI). Furthermore, aerosol particles can act as seeds for cloud droplets and, hence, changes in aerosol composition and concentration can alter cloud properties, affecting Earth's radiative balance indirectly (aerosol-cloud interactions; ACI).  Aerosols can also affect local meteorological dynamics. For instance, absorbing aerosol particles can alter atmospheric stability in the troposphere due to local heating (Johnson et al, 2019)."

*L72-73: Cannot find in any of the given references that 'emissions from New Delhi' significantly contribute to the ATAL. Please clarify.*

This statement was slightly misleading, as Fairlie et al. (2020) referred to Indian subcontinent and Northern India as significant emission sources contributing to the ATAL.

Updated the sentence at lines 72-73 from

"emissions originating from New Delhi contribute significantly to the Asian Tropopause Aerosol layer (ATAL)" to

"emissions originating from Indian subcontinent contribute significantly to the Asian Tropopause Aerosol layer (ATAL)"

*L94: HAM 'threats' the chemical combounds..?*

Many thanks for pointing out this typing error.

We have updated the sentence at line 94 from

"HAM threats the chemical compounds" to

"HAM treats the chemical compounds"

*L158: Why do OC emissions increase by 2030 in CLE scenario while all others show a reduction?*

The principal reason for the different trends in different aerosol compound emissions are differences in the relative contribution of the total emissions from each emission sector. These relative contributions may also change differently based on the burnt fuels and technologies used. In the CLE scenario for the year 2030, the *global* OC emissions from waste sector increase approximately by 900 kt (+44 %) when compared to 2015 levels. Furthermore, emissions due to agricultural waste burning increase by ~200 kt (+10%) compared to year 2015 emissions, and the emissions from industry and shipping are also projected to increase by a small portion. These changes are almost as large as emission reductions in the domestic (~ -960 kt) and traffic

(~ -230 kt) sectors. The net change in global anthropogenic OC emissions is a small decrease (~ -0.07 %) in the year 2030 compared to the year 2015.

For the *area surrounding New Delhi*, the increasing OC emissions from the waste (~ +90 kt, +62%) and industry (+11kt, +83%) sectors counterbalance some of the emission reductions cuts from the domestic (-38kt) and traffic (-18kt) sectors. Therefore, there is an increase in the net OC emissions in the area surrounding New Delhi. For BC and SO2, the reductions in other sectors (domestic and traffic for BC and energy sector for SO2) are large enough to outweigh the increasing emissions from the waste sector. That is why the 2030 CLE emissions for BC and SO2 are less than in 2015.

We have altered the sentence at line 155 from
"while the global OC emissions are projected to stay almost constant ($\sim -0.07$ %)" to
"The change in net global OC emissions is almost negligible (~ -0.07 %) due to increasing emissions in the waste and agricultural waste burning sectors, which compensate emission reductions projected in the domestic and traffic sectors."

And also modified the sentence at line 157 from
"The OC emissions are an exception as they increase by 7 % in 2030 compared to 2015." to
"The OC emissions are projected to increase for the New Delhi surrounding by 7 % in the year 2030 compared to the year 2015. This is due to increasing emissions from the waste and industry sectors, despite emission cuts projected for the domestic and traffic sectors."

*L267: What is '2D yearly mean value'?*
By 2D, we meant that the yearly mean values were calculated separately for each grid box.

We changed the sentence at line 267 from
"For both $RF_{ARI}$ and ERF, we first calculated 2D yearly mean values using $RE_{ARI}$ and the net TOA radiative flux, respectively." to
"In order to compute $RF_{ARI}$ and ERF, we first computed 2D yearly mean fields of $RE_{ARI}$ and the net TOA radiative flux, respectively, which were calculated as yearly mean values for each separate grid box."

*L356: If that is the case, what is the significance of feature importance values?*

The feature importance values describe which input features reduced the modeling error the most in the training phase. Therefore, the importance values reveal information about the RF algorithm priorities during the training. However, the feature importances do not necessarily describe the RF model output sensitivities to input features.

To describe this better, we have added a sentence after line 357:
"The feature importance values indicate the contribution of a feature to the total reduction in the error criterion. However, importance values do not reveal the sensitivity of the RF model to

specific input features."

*L385: This __ somewhat?*

We have modified this sentence to clarify the main point, I.e., that the strong negative $RF_{ARI}$ values were not expected when all three aerosol species are reduced simultaneously.

Changed the statement at line 385 from
"This somewhat surprising result is due to ..." to
"This result is somewhat surprising as both of the scenarios, CLE_2030 and MITIG_2030, included reductions in both absorbing and scattering aerosols. The net negative $RF_{ARI}$ for India results from ..."

*L395-398: Confusing. The aerosol loading in MITIG_2030 is supposed to be lower than the CLE_2030, then how RF in MITIG_2030 is more negative than CLE_2030 at the Himalayan foothills. Bright background due to strong haze is expected to be more in CLE_2030.*

The $RF_{ARI}$ describes the change in the aerosol radiative effect between perturbed and reference scenarios. MITIG_2030 $RF_{ARI}$ is more negative than CLE_2030 $RF_{ARI}$ since there is less absorbing aerosol (BC) in MITIG_2030 than in CLE_2030. It is true that the pollution haze is expected to be stronger in CLE_2030 than in MITIG_2030. However, as we see from Figures S3a and b, the atmospheric absorption is significantly less in MITIG_2030 than in CLE_2030. This indicates that the absorption due to BC is the dominating effect in this area, and therefore the $RF_{ARI}$ is more negative in MITIG_2030 than in CLE_2030.

We have added the following sentences after line 398:
"The $RF_{ARI}$ is more negative for the MITIG_2030 simulation, even though the haze layer due to anthropogenic aerosols is stronger in CLE_2030. This is because BC is reduced more heavily in MITIG_2030, which leads to stronger contributions to $RF_{ARI}$ than the changes in the background aerosol layer."

*L434-436: cannot understand. Do you mean that the CDNC burden was more in CLE_2030 scenario?*

Exactly.

To make this point clear, we rewrote the sentence at lines 434-346, from
"When comparing CLE_2030 and PRES, the changes in aerosol concentrations translate surprisingly into a very small increase in the CDNC burden over most of the India (not shown), which would indicate that there is more cloud droplets to scatter SW radiation." to
"When comparing CLE_2030 and PRES, the changes in aerosol concentrations translate surprisingly into a very small increase in the cloud droplet number concentration (CDNC)

burden over most of India (not shown), i.e. the mean CDNC burden values are higher in CLE_2030 than in PRES for large areas over India. This would indicate that there are more cloud droplets to scatter SW radiation."

*L435: Expand CDNC in the manuscript.*
Thanks for pointing out this deficiency.

We have clarified the abbreviation by modifying line 435 from
"in the CDNC burden" to "in the cloud droplet number concentration (CDNC) burden"

In addition, we described the term "burden" in Section 2.6 by adding in the Table 3 caption following:
"Burden indicates vertically integrated concentration values."

*Conclusions: This section can be renamed as summary and conclusions by adding the significant findings of the study as bullet points.*

Many thanks for the suggestion, we have renamed Section 4 and have added a very short summarizing statement to the end of this section. We, however, felt that a bullet-pointed list would not fit the style of the journal and instead opted for giving the summary in plain text style.

We have added after line 532 the following:
"To summarize, we obtained greatly improved seasonal trends for surface PM2.5 when applying machine learning downscaling to ECHAM-HAMMOZ data for New Delhi. Stringent, global aerosol mitigation resulted in improved air quality in New Delhi and negative radiative forcing for most of India. Organic carbon emissions had a stronger influence on local air quality, whereas black carbon emissions contributed more to the radiative effects."

[revised manuscript text omitted]

---

## Author Response (AR2)

**Responses to the referee comments:**
**Assessing the climate and air quality effects of future aerosol mitigation in India using a global climate model combined with statistical downscaling**

*Miinalainen, T., Kokkola, H., Lipponen, A., Hyvärinen, A.-P., Soni, V. K., Lehtinen, K. E. J., and Kühn, T.: Assessing the climate and air quality effects of future aerosol mitigation in India using a global climate model combined with statistical downscaling, Atmos. Chem. Phys. Discuss. [preprint], https://doi.org/10.5194/acp-2022-513, in review, 2022.*

We thank Anonymous Referees #2 and #3 and the Editor for the good comments that helped us to improve our manuscript. Our responses are written below each comment separately. The referee comments are marked with *yellow color and italic*, and the author replies are marked with gray color. The original manuscript text is marked with pink color, and updated text with dark magenta. The line numbers refer to the 1[st] revised, submitted version of the manuscript which was peer-reviewed.

**Replies to the comments made by the Editor:**

*"[...] In addition to these, I found some very small technical questions. Firstly, what are Aprc and Aprl in Figure 1 - I couldn't quite see which input variable from Table 3. Are they the precipitation variables? Perhaps putting that abbreviation in Table 3 could be helpful. The others were clear to me. Also, in Figure 1a there isn't a dark blue line in the legend, rather it shows up gray to me. Finally, winter and summer months are defined in line 473, but you talk about winter and summer before that. It may help to define the months the first time you discuss the seasons."*

We thank the Editor for carefully reading our manuscript and for the suggestions how to improve the readability of our manuscript. We have made the following changes:

*Table 1 is now updated as suggested to describe the abbreviations that will be used later in Figures 1 and 2.

*Figure 1a has been updated by changing the shade of blue for the lines that represent the individual RF-corrected PM2.5 values.

*We have modified the manuscript text to mention explicitly the months that we are referring to when discussing summer and winter seasons.

Furthermore, we noticed that there was some missing information about the modeling setup used for the global model simulations. We added the following sentences after line 116:

"In addition, we used an additional setup where the emitted BC was assumed to get directly internally mixed with sulfate (Holopainen et al., 2020). Therefore, BC-containing particles were modeled to be more soluble already at the time of their emission."

\* \* \* \* \* \* \* \* \* \* \* \* \* \* \* \* \* \* \* \* \* \* \* \* \* \* \* \* \* \* \* \* \* \* \* \* \* \* \* \* \* \* \* \* \* \* \* \* \* \* \* \* \* \* \* \* \* \* \* \* \* \* \* \* \* \* \* \* \* \* \* \* \* \* \* \* \* \* \* \* \* \* \*

**Replies to the comments made by the Anonymous Referee #2:**

*The manuscript has improved substantially after the revision and the effort of the authors in addressing all the aspects of the review is appreciable. The second part of the work which examines the impact of future emission scenarios in terms of radiative forcing is now well-discussed with proper references. While all seem perfect, a few concerns/queries regarding the ML technique based on the responses to the reviewer comments are still pending which are briefed below.*

We thank Anonymous Referee #2 for the comprehensive review and for the valuable comments. The more detailed replies are listed under each comment separately

*• Why this model is 'not at all sensitive or unaffected' to the input parameters (Fig./Table1 for the 2nd reviewer) as well as hyper-parameters (Fig. 2 for the 2nd reviewer)? The correlation coefficient values are expected to change if the parameters have an association with the target values, but there is no change. According to the 2nd figure, the model appears insensitive to the max depth feature. The model can acquire all the necessary information by the initial split itself, which seems unrealistic.*

We repeated the test that was conducted for the previous paper revision and where we altered the maximum depth parameter. This time we changed the max_depth parameter from 1 to 5, and used again the year 2020 data (i.e., outside of training and testing data). The error statistics for the repeated test are listed in the table below.

| max_depth | 1 | 2 | 3 | 4 | 5 |
|---|---|---|---|---|---|
| RMSE ($\mu$g/m$^3$) | 58.05 | 55.03 | 53.75 | 53.31 | 53.01 |
| MRE (%) | 71.81 | 66.58 | 63.49 | 62.81 | 61.86 |
| MAE ($\mu$g/m$^3$) | 45.91 | 42.97 | 41.55 | 41.08 | 40.64 |
| $R^2$ | 0.46 | 0.51 | 0.53 | 0.54 | 0.55 |
| Pearson correlation | 0.80 | 0.80 | 0.80 | 0.80 | 0.80 |

The results clearly show that the model improves with max_depth being increased. At some point the model's maximum capacity is reached and accuracy metrics do not improve anymore. This is the case in the earlier reply to referee and the accuracy metric was not changing notably

regadless of e.g. different max_depths. In many machine learning models, such as neural networks, overfitting may be a significant problem. If the capacity of the machine learning model is increased too much it may lead to overfitting and the accuracy metrics of the test data decreases. Here we have shown that our Random Forest model does not suffer from overfitting and produces the best possible results based on our training data given the capacity of the model is large enough. In our case, the capacity leading to optimal performance is achieved already with a relatively low number of max_depth.

Note that here the error statistics differ from the error statistics presented in the manuscript. This is because we used here data from the year 2020, whereas in the manuscript the testing data is from the years 2018 and 2019.

Furthermore, error statistics are calculated based on the average over all corrections and daily average values of the measurement stations. This means that slight changes in one RF-correction do not affect the outcome as heavily as it would be the case if analyzing the outcome of one single RF model. All 31 RF models use the same hyperparameters, and there is no individual tuning for the RF model parameters.

- *L467-469 in the track-changed version: These two sentences are mutually contradicting. As per the first sentence, 'the feature importance value indicates the contribution of a feature to the total reduction in the error criterion', then, what is meant by the second sentence- 'Feature importance values do not reveal the sensitivity of the RF model to specific input feature'? Please clarify/modify.*

Thank you for bringing to our knowledge that this part was not clear in the manuscript text. The error criteria used in our model is the mean squared error, and the feature importance values indicate the contribution of a feature to the total reduction in the error. However, the reduction in error does not necessarily describe the total effect of a feature on the model prediction. Some features might affect, for instance, summertime values by increasing the magnitude of the estimate, but still the net error might be of the same magnitude as without the feature.
We have updated in the lines 413-414 the sentence from:
"However, importance values do not reveal the sensitivity of the RF model to specific input features." to
"However, importance values do not reveal the sensitivity of the RF model to specific input features as the reduction in error does not indicate directly how much a feature affects the RF model output trends and magnitude."

- *How important/necessary is normalization in Random Forest which is a tree-based model and not a 'Neural Network Model'?*

Input normalization is not important in training a Random Forest model. The splitting of the data in the construction of the regression trees is based on the ordering of the input variable.

The input variable ordering is not affected by the normalization and thus the input normalization is not important in Random Forests. Therefore, we did not normalize the input data used in our RF corrections. However, the feature importance values, which are an output of the RF training procedure, were normalized to be in a scale from 0 to 1.

*• When the authors state that bagging is not used, does it mean that the entire dataset is fed into the model rather than in short batches? Not able to understand the term 'Bootstrap bagging'.*

Since the bagging is not used, the whole data set is used for building each tree. However, the parameter "max_features" controls the number of features used in each split. This way, there are differences in the trees and variation in the output of different trees. Bagging refers to conducting model training with bootstrap samples many times. As mentioned in the previous revision comments, bootstrap sampling was not used in our analysis.

\* \* \* \* \* \* \* \* \* \* \* \* \* \* \* \* \* \* \* \* \* \* \* \* \* \* \* \* \* \* \* \* \* \* \* \* \* \* \* \* \* \* \* \* \* \* \* \* \* \* \* \* \* \* \* \* \* \* \* \* \* \* \* \* \* \* \* \* \* \* \* \* \* \* \* \* \* \*

**Replies to the comments made by the #Anonymous referee 3:**

*General remarks: The manuscript deals with aerosol near surface concentrations from a GCM, downscaled with a random forest approach over Dehli (India). The downscaling correction substantially improves the GCM performance in much better agreement with the observations. Therefore, the authors show the potential of the method. After a few minor corrections, the paper should be accepted for publication in my opinion.*

We thank Anonymous Referee #3 for the excellent suggestions and for dedicating time to go through our manuscript. The detailed answers are listed below under each comment.

*Content:*
*It is unclear how the station values are obtained from the GCM simulations, i.e. is it nearest neighbour or linear or cubic interpolation. How are potential altitude misrepresentations considered in determining the training values from the GCM at the station locations?*

Thank you for making us aware that this aspect was not mentioned in the manuscript text. The data for New Delhi from ECHAM-HAMMOZ was retrieved by using nearest neighbor mapping. In practice, this was done by using the CDO program method "remapnn" and using the lowest model level output data. It is true that for some of the stations, the altitude of a station might be slightly higher than what the lowest model level represents. However, we estimated that in New Delhi the stations are located mostly near ground level, and this would not cause a significant error in our modeling studies. Furthermore, the aim of this study was to obtain one average PM2.5 value for the New Delhi urban region, which would be representative of whole

area. Therefore, in order to minimize the order of complexity of the method, all RF input feature data was retrieved from the same vertical model layer, and for the same latitude and longitude coordinates.

We have modified the sentence in line 272 to be from:

"For some input features, we used values representing one grid box surrounding the New Delhi region (point)." to
"For some input features, we used values representing one grid box surrounding the New Delhi region (point). These were retrieved from ECHAM-HAMMOZ data by using nearest neighbor interpolation."

*How can the R^2 value in table 4 be negative for the uncorrected output?*
The R-squared value is computed as $R^2 = 1 - SS_{res} / SS_{tot}$ , where $SS_{res}$ is the sum of squared residual errors between the modelled and measured values, and $SS_{tot}$ is the sum of variance in the modelled data. If the modelled values do not follow the trends of measured data, the R-squared values can be negative. The R-squared value does not represent the squared Pearson correlation value, though this might be understood from Table 4 as we had Pearson correlation also marked with the letter R. Therefore, we have modified Table 4 to explicitly mention Pearson correlation and removed the abbreviation "R" to avoid confusion with the R-squared.

*Can it be estimated, how large is the impact of the ML based correction on the surface concentrations on the total forcing? Even though there is the difficulty of estimating the effect of near surface concentrations on TOA forcing, the question whether the surface values have a large impact on the total forcing and therefore the correction would be also beneficial for the radiative forcing corrections. A short discussion on this topic should be added to the manuscript.*

This was a good question. In principle, what the referee suggests could be achieved, but there are several to keep in mind: The corrected PM2.5 values represent very local PM2.5 values for New Delhi urban region, and do not represent the whole grid box. Therefore, to make better radiative forcing predictions for one grid box, calculations would have to be done on a sub-grid scale, including much more observational data spanning the entire grid box (for ECHAM-HAMMOZ the mentioned 2°x2°). However, TOA radiative fluxes are analyzed for larger areas, and not only one grid box as it would not be very representative if considering the energy fluxes over longer term periods. Therefore, the estimation of changes in radiative fluxes for larger areas would require even more station PM2.5 data, from various locations in India, and this would require separate RF models for each of the stations. In addition, as radiative forcing calculations are performed online, the RF correction would need to be conducted dynamically as well, i.e., during the ECHAM-HAMMOZ simulation, in order for the surface concentrations to affect TOA radiative fluxes. Furthermore, we here only correct for PM2.5, which is the integral over the aerosol size distribution up to 2.5 μm. For radiation calculations, on the other hand, the aerosol size information must be preserved, as ARI strongly varies with aerosol particle size.

This exercise, though very interesting, would be computationally quite demanding, and therefore is out of the scope of this study.

We have added after Line 596 the following:
"As a continuation to this study, one could apply the downscaling method described here during the ECHAM-HAMMOZ simulation (instead of after, as it was done here) and for larger areas. With some extensions which address additional aspects like, e.g., information about aerosol size, this may allow the bias correction to also affect the computation of other aerosol impacts, like radiative fluxes. However, such a dynamical approach for larger areas would also require a much larger spatial coverage of observational data, which would make the model computationally more demanding. Furthermore, without proper evaluation, such a model extension might introduce further uncertainties to the radiative forcing estimation."

*line 148: specie -> species*
Thanks for the suggestion, we have corrected this typing error. We have changed in line 148:
"the values for each grid box and emission specie from year 2015 to year 2020." to
"the values for each grid box and emission species from year 2015 to year 2020."